# tRNA m¹A modification regulate HSC maintenance and self-renewal via mTORC1 signaling

Hongna Zuo [1,2], Aiwei Wu [1,2], Mingwei Wang [1], Liquan Hong[1] & Hu Wang [1] ✉

Haematopoietic stem cells (HSCs) possess unique physiological adaptations to sustain blood cell production and cope with stress responses throughout life. To maintain these adaptations, HSCs rely on maintaining a tightly controlled protein translation rate. However, the mechanism of how HSCs regulate protein translation remains to be fully elucidated. In this study, we investigate the role of transfer RNA (tRNA) m¹A58 'writer' proteins TRMT6 and TRMT61A in regulating HSCs function. *Trmt6* deletion promoted HSC proliferation through aberrant activation of mTORC1 signaling. TRMT6-deficient HSCs exhibited an impaired self-renewal ability in competitive transplantation assay. Mechanistically, single cell RNA-seq analysis reveals that the mTORC1 signaling pathway is highly upregulated in HSC-enriched cell populations after *Trmt6* deletion. m¹A-tRNA-seq and Western blot analysis suggest that TRMT6 promotes methylation modification of specific tRNA and expression of TSC1, fine-tuning mTORC1 signaling levels. Furthermore, Pharmacological inhibition of the mTORC1 pathway rescued functional defect in TRMT6-deficient HSCs. To our knowledge, this study is the first to elucidate a mechanism by which TRMT6-TRMT61A complex-mediated tRNA-m¹A58 modification regulates HSC homeostasis.

Haematopoietic stem cells (HSCs) are a rare population of bone marrow-resident adult stem cells that have an extensive capacity for self-renewal and multilineage differentiation and maintain lifelong blood cell production in the body[1]. In order to maintain lifelong stem cell function and cope with various internal and external stresses such as replicative, metabolic, oxidative and genotoxic stresses[2–13], HSCs have acquired various adaptations that can help maintain their function. While considerable insights have been generated through studies targeting specific genes or pathways in HSCs[4,6,9,14–21], the mechanisms underlying the full range of adaptations found in HSCs are still not fully understood.

HSCs undergo rapid and profound changes to break out of quiescence, followed by massive HSC clonal expansion and differentiation, which is essential for adequate haematopoietic reconstitution. HSCs meet this enormous bioenergetic and biosynthetic demand by rapidly increasing protein synthesis at the transcriptional, post-transcriptional and translational levels[22–26]. mTOR pathway plays an important role in these processes and has been identified as a key factor in regulating HSC function[27]. Disruption of the mTOR pathway in adult mouse HSCs by the introduction of loss-of-function alleles of Pten or Tsc1 (both negative regulators of mTOR)[15,17,28–30] or gain-of-function mutations in the mTOR activators RHEB2 or AKT leads to depletion of HSC recycling and long-term reconstitution activity[24,27]. Furthermore, pharmacological inhibition of mTOR with rapamycin restored haematopoietic stem cell activity in these models[27,31]. These studies strongly suggest that mTOR activity

[1]Zhejiang Key Laboratory of Medical Epigenetics, School of Basic Medical Sciences, The Third People's Hospital of Deqing, Department of Cardiology, Affiliated Hospital of Hangzhou Normal University, Hangzhou Normal University, Hangzhou 311121, China. [2]These authors contributed equally: Hongna Zuo, Aiwei Wu. ✉e-mail: wanghu19860315@163.com

plays an important role in haematopoietic stem cell function. However, the regulation of the mTOR pathway at the level of translational regulation has been less well studied.

It has long been thought that tRNAs influence translation through their structure and interaction with the corresponding mRNA codons[32–34]. Their regulation is closely linked to different tRNA chemical modifications, which have been the subject of numerous in vitro studies[35]. Mammalian tRNAs are the most highly modified RNA molecules in the cell[35]. On average, they contain 14 modified nucleotides per molecule[36]. tRNA modifications have multiple roles involving control, decoding and so on[34]. One of the evolutionarily conserved epitope marks is N[1]-methyladenosine (m[1]A), which is normally found at position 58 of tRNA and is catalyzed by the tRNA methyltransferases TRMT61A and TRMT6[33,37–39]. TRMT6 and TRMT61A-mediated tRNA-m[1]A58 modification has been reported to enhance translation initiation and elongation[34,36,40–45].

However, the in vivo biological function of the tRNA-m[1]A58 methyltransferases TRMT6/TRMT61A in HSCs remains completely unknown. Recently, He HQ et al. showed a new molecular mechanism which is over-expression of TRMT6-TRMT61A complex promotes HSC aging through independent role of tRNA-m[1]A58 modification[46]. Here, we observed that specific deletion of *Trmt6* in HSCs resulted in short-term aberrant expansion of HSCs, and a significant and substantial decrease in self-renewal capacity. Mechanistically, TRMT6-mediated tRNA-m[1]A58 is mounted on the TSC1 protein to keep mTOR in the proper state to rapidly initiate mTOR signaling when necessary, and it maintains haematopoietic stem cell stemness and stability. In conclusion, these results suggest that the tRNA-m[1]A58 methyltransferases TRMT6/TRMT61A acts as a translational checkpoint for the mTOR pathway and constitute an important mechanism responsible for maintaining HSC homeostasis and the rapid synthesis of specific key functional proteins to promote HSC differentiation and self-renewal.

## Results

### *Trmt6* loss leads to phenotypic HSC expansion and Long-term repopulation capacity decline

Before investigating the biological roles of *Trmt6* and *Trmt61a* in haematopoietic system, we initially assessed the expression of *Trmt6* and *Trmt61a* genes across multiple haematopoietic cell populations utilizing publicly available single-cell transcriptomic data[47]. *Trmt6* and *Trmt61a* are broadly expressed in haematopoietic system (Figs. S1a and g). Notably, we observed a consistent upregulation of *Trmt6* but not *Trmt61a* in donor-derived bone marrow cells derived several days post-transplantation of HSCs[47] (Figs. S1b and h). Treatment of HSCs with 5-Fu[48], ionizing radiation[49], or LPS[50] resulted in upregulation of *Trmt6* but not *Trmt61a* at 1-3 days (Figs. S1c–S1e and S1i–S1k), while in aged HSCs the expression of *Trmt6* slightly decreased[51] (Fig. S1f), indicating that *Trmt6* and *Trmt61a* may play a role in homeostasis and regeneration of HSCs.

To understand the functional role of *Trmt6* in haematopoietic system, we generated *Mx1-Cre;Trmt6[fl/fl]* (*Trmt6[-/-]*) mice and confirmed the deletion efficiency of *Trmt6* (Figs. 1a, b and S2a). Although the size and weight of *Trmt6[-/-]* spleen were comparable with that of wild-type mice, a reduction in frequency of B lymphocytes and an increase in the frequency of myeloid cells were observed in spleen of *Trmt6[-/-]* mice (Figs. S2b–d). In peripheral blood (PB) of *Trmt6[-/-]* mice, the reduction in frequency of B lymphocytes and the increase in the frequency of myeloid and T cells were more noticeable (Fig. S2e). Furthermore, complete blood counts revealed a significant decrease in the number of white blood cells and lymphocytes and a significant increase in the number of platelets and neutrophils in *Trmt6[-/-]* mice (Figs. S2f–i). Those results suggested an abnormal lineage output in *Trmt6[-/-]* mice.

To formally and functionally investigate the effects of *Trmt6* loss in the haematopoietic system, we evaluated the immunophenotypic composition of the HSPC compartments in cohorts of *Trmt6[+/+]* and

*Trmt6[-/-]* mice. Flow cytometry analysis showed a ~ 4-fold increase in the frequency of CD150[+]CD48[-]FlK2[-] LSK cells (LT-HSCs) and progenitors (MPP, GMP, CMP, and MEP) in *Trmt6[-/-]* bone marrow at 1week after polyI:C (Fig. 1c–e). Taken together, *Trmt6* deletion causes a dramatic increase in the numbers of HSCs.

To acquire further insight into the expansion of HSC pools after *Trmt6* deletion, we conducted cell cycle analysis on LT-HSCs and LSK cells. Ki67 staining revealed a significantly lower frequency of $G_0$ cells (Ki67[-]) in *Trmt6[-/-]* LT-HSCs and LSK cells than that in *Trmt6[+/+]* cells (80% in *Trmt6[+/+]* *vs.* 58% in *Trmt6[-/-]* LT-HSCs) (Figs. 1f, g and S3a). Cycling HSCs are more sensitive to 5-Fluorouracil (5-Fu) cytotoxicity, while quiescent HSCs remain viable after 5-Fu treatment[52]. *Trmt6[-/-]* mice challenged with sequential 5-Fu treatment died significantly earlier than the *Trmt6[+/+]* controls (Figs. S3b and c), indicating that *Trmt6*-null HSCs were less quiescent.

Aberrantly enhanced proliferation often consequentially exhausts HSCs and affects their function in haematopoietic reconstitution[8,9,24,53–55]. A significant reduction of BM cells in *Trmt6[-/-]* mice was found by examining the total number of bone marrow cells (Fig. S3d). To further observe the HSC depletion phenomenon caused by TRMT6 deficiency, flow cytometry analysis showed that the frequency of LT-hematopoietic stem and progenitor cells (MPP2, MPP3, MPP4) in *Trmt6[-/-]* bone marrow was significantly decreased 4 weeks after injection of PolyI:C (Fig. S3e), and at the same time, compared with control mice, the survival rate of *Trmt6[-/-]* mice was significantly decreased (Fig. S3h), and even less for stress induced by 5-Fu (Figs. S3b and c) and 4 Gy irradiation (Figs. S3i and j). Collectively, these data suggest that *Trmt6* depletion has a significant impact on hematopoiesis.

We hypothesize that although higher in number, the functionality of *Trmt6*-null HSCs might be exhausted faster upon transplantation or stresses. To this end, we conducted HSC competitive transplantation experiments. Surprisingly, in comparison with *Trmt6[+/+]* HSCs, *Trmt6[-/-]* HSCs exhibited a ~ 10-fold reduction in their long-term repopulating ability followed by PB and LT-HSC chimerism analysis (Figs. S3f and g).

To exclude the potential impact of TRMT6 deficiency on HSC homing upon transplantation, *Trmt6[+/+]* Mx1-Cre or *Trmt6[fl/fl]* Mx1-Cre BM cells were first transplanted without inducing *Trmt6* deletion. The recipient mice were then injected with polyI:C at 6 weeks after transplantation to induce *Trmt6* deletion, followed by PB chimerism analysis and secondary transplantation (Fig. 1h). To this end, we conducted competitive transplantation experiments by injecting $1 \times 10^6$ *Trmt6[+/+]* or *Trmt6[-/-]* BM cells (CD45.2 background) into lethally irradiated congenic-recipient mice (CD45.1) along with $1 \times 10^6$ competitive BM cells (CD45.1), followed by secondary transplantation (Fig. 1h). In contrast to *Trmt6[+/+]* cells, which gave rise to stable long-term multi-lineage reconstitution in the recipient mice, *Trmt6[-/-]* LT-HSCs and PB exhibited a ~ 10-fold reduction in their long-term repopulation ability at 20 weeks post transplantation (Figs. 1i, j and S3k–m). The analysis of recipient mice revealed a significant decrease in the frequency of donor-derived PB and LT-HSC cells 16 weeks after the secondary transplantation (Figs. 1k, l and S3n–p), suggesting that *Trmt6* deletion produces a cell-autonomous functional defect in HSCs. Altogether, these data indicate that TRMT6 depletion impaired the long-term multi-lineage repopulation activity and self-renewal capacity of HSCs in a cell intrinsic manner.

### Single-cell RNA sequencing (scRNA-seq) revealed an activated mTORC1 signaling pathway in *Trmt6* deficient HSC populations

To elucidate the molecular mechanism of *Trmt6* in regulating haematopoiesis, we performed single-cell RNA sequencing (scRNA-seq) on sorted HSPCs (LSK) cells isolated from bone marrow (BM) of young adult *Trmt6[+/+]* and *Trmt6[-/-]* mice (Fig. 2a and S2j). In total, 8258 out of 8869 and 8599 out of 9260 cells from *Trmt6[+/+]* and *Trmt6[-/-]* BM passed

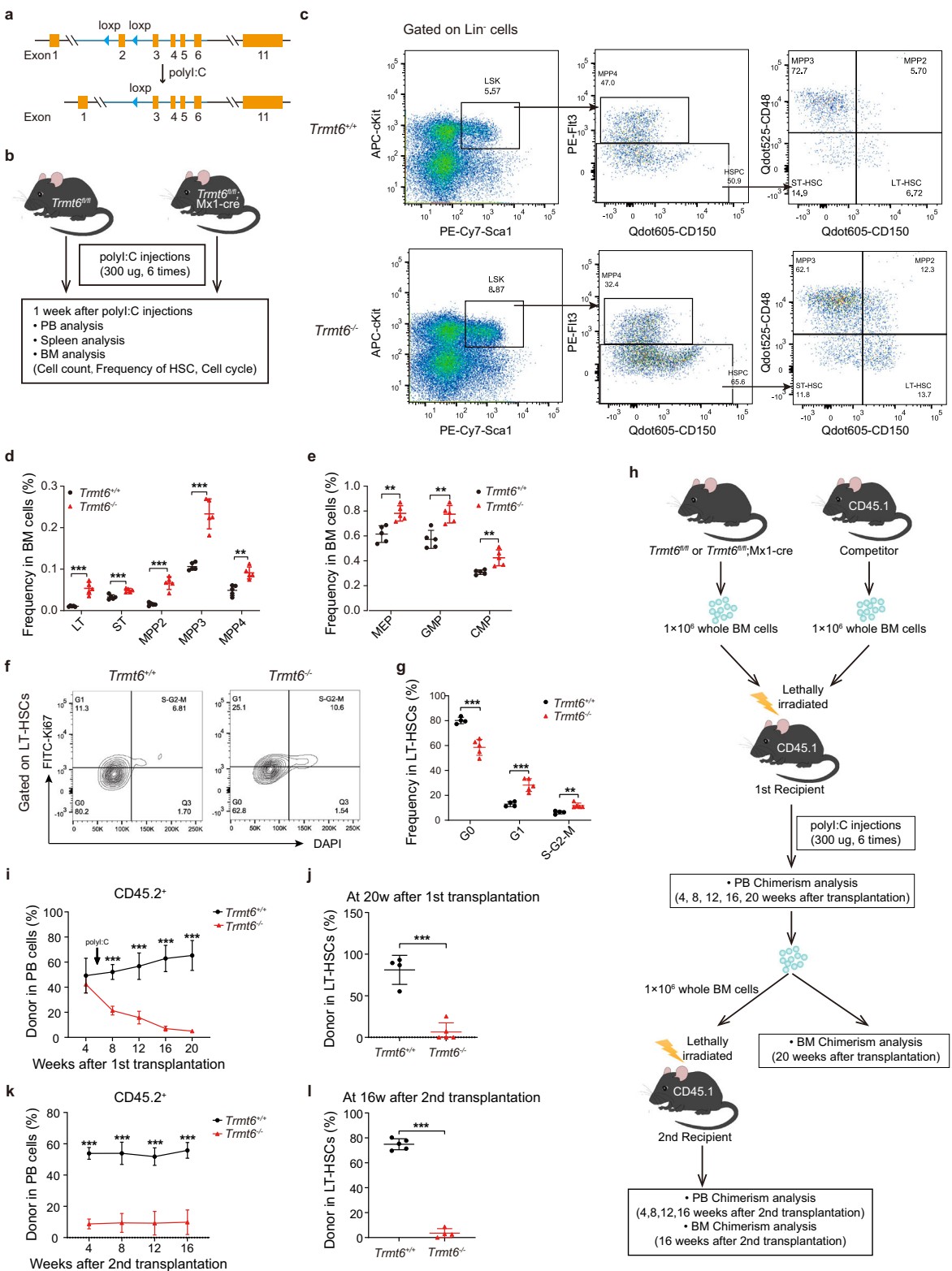

quality control, respectively. Cells were annotated using SingleR package by comparing transcriptome between our and Nestorowa et al. data[56] (Figs. 2b, c and S4a). *Procr/Pdzk1ip1*, *Myct1*, *Dntt*, *Ctsg* and *Car1* were specifically and highly expressed genes in the predicted long-term HSCs (LT-HSCs), MPPs, LMPPs, CMPs, GMPs and megakaryocyte-erythroid progenitors (MEPs), respectively

(Figs. S4b and c). Dramatic changes of cell type composition were observed in *Trmt6*[-/-] HSPCs, specifically, the fraction of LT-HSCs increased, along with increased fraction of multipotent progenitor 2 (MPP2) population and decreased fraction of other cell types (Fig. 2d). *Trmt6*[-/-] HSPCs exhibited higher proliferation score, and the percentage of cells with S and G2/M cell cycle signature elevated in *Trmt6*[-/-]

**Fig. 1 | *Trmt6* is essential for the repopulating potential of HSCs. a** Targeting strategy to generate the *Trmt6* conditional knockout (cKO) mouse. **b** Experimental scheme for *Trmt6⁻/⁻* and *Trmt6⁺/⁺* hematopoiesis analysis. **c** FACS analysis of LT-HSCs (CD150⁺CD48⁻Flk2⁻Lin⁻c-Kit⁺Sca-1⁺), ST-HSCs (CD150⁻CD48⁻Flk2⁻Lin⁻c-Kit⁺Sca-1⁺), and MPPs in *Trmt6⁻/⁻* and *Trmt6⁺/⁺* BM cells. Representative FACS profiles are shown. $n = 5$ mice per genotype. **d** Frequency of LT, ST, MPP2, MPP3 and MPP4 in BM cells are shown. $n = 5$ mice per genotype. **e** Frequency of MEP, GMP and CMP in BM cells are shown. $n = 5$ mice per genotype. **f** Cell cycle analysis of *Trmt6⁻/⁻* and *Trmt6⁺/⁺* LT-HSCs. Representative FACS profiles of Ki67 antibody and DAPI staining are shown. **g** Frequency of the cell cycle distribution in *Trmt6⁻/⁻* and *Trmt6⁺/⁺* LT-HSCs are shown. $n = 5$ mice per genotype. **h** Experimental schematic for serial competitive transplantation with *Trmt6⁻/⁻* and *Trmt6⁺/⁺* BM cells. **i** Percentage of donor-derived PB cells at the indicated time points in the first competitive transplantation. $n = 6$ mice per genotype. **j** Percentage of donor-derived LT-HSCs 20 weeks after first transplantation. $n = 6$ mice per group. **k** Percentage of donor-derived PB cells at the indicated time points in the second competitive transplantation. $n = 6$ mice per group. **l** Percentage of donor-derived LT-HSCs 16 weeks after secondary transplantation. $n = 6$ mice per group. Data represent the mean ± SD from three independent experiments. *, $P < 0.05$; **, $P < 0.01$; ***, $P < 0.001$. For all the above statistics, $P$ values were obtained using unpaired parametric two-tailed t-test. Exact $P$ values are provided as Source Data.

HSCs and MPPs (Figs. 2e, f), suggesting a disturbed quiescence-proliferation transition in *Trmt6⁻/⁻* HSCs.

To explore the mechanism whereby *Trmt6* controls the quiescence-proliferation transition, we performed differential expression analysis for each cell type. In each cell type except for MEPs (where only 3 genes were dysregulated), hundreds of genes showed dysregulation following *Trmt6* depletion (adjusted $P < 0.05$ and | $\log_2 FC | > 0.25$), with some of them exhibiting a larger change in expression (adjusted $P < 0.05$ and $|\log_2 FC| > 1$) (Fig. S4d). In the case of LT-HSCs, 411 genes were found to be differentially expressed upon *Trmt6* depletion, with 325 genes upregulated and 86 genes down-regulated. Out of these, 36 genes were specifically differentially expressed in LT-HSCs, while the remaining 375 genes showed dysregulation in downstream haematopoietic cells. This observation suggested that the impact of *Trmt6* depletion is long-lasting. Gene set enrichment analysis (GSEA) of gene expression changes in *Trmt6⁻/⁻* LT-HSCs showed upregulated proliferation signature and downregulated quiescence signature (Fig. 2g). Cell-cell communication analysis showed that HSCs interacted more frequently with others in *Trmt6⁻/⁻* mice than those in *Trmt6⁺/⁺* mice (Figs. S4e and f). Above results indicated *Trmt6* depletion, to some extent, deprived the restriction of quiescence-proliferation transition. Further GSEA reveals more active expression of several hallmark gene sets *Trmt6⁻/⁻* HSPCs, including mTORC1 signalling, unfolded protein response, MYC targets, oxidative phosphorylation, and glycolysis (Fig. 2g). Additional gene set variation analysis (GSVA) also showed significantly higher expression of mTORC1 signalling genes in *Trmt6⁻/⁻* LT-HSCs than that in *Trmt6⁺/⁺* LT-HSCs (Wilcoxon rank sum test, $P < 2.2e-16$) (Fig. 2h). It has long been established that mTORC1 signalling governs the quiescence of HSCs[57,58]. therefore, *Trmt6* likely controls the quiescence-proliferation transition through regulating mTORC1 signalling. Consistent with the observed lineage-bias of haematopoietic system in *Trmt6⁻/⁻* mice and the known role of mTOR signalling pathway as the central hub for HSC fate coordination[27], pseudotime analysis showed a prone-to-myeloid transcriptomic change of HSCs in *Trmt6⁻/⁻* mice (Fig. 2i), and the transcriptomic shift leading to lineage-bias likely happened in early differentiation stage, namely, in LT-HSCs (Fig. 2j). In all, single-cell transcriptomic profiling of HSPC populations supports a prominent role for *Trmt6* in HSC quiescence.

## tRNA-m¹A58 modification is required for efficient *Tsc1* mRNA translation and mTORC1 signaling regulation

Previous studies have shown that the mammalian target of rapamycin complex 1 (mTORC1) is critical for cell metabolism, cell growth and cell cycle progression in HSCs. Therefore, we first examined the activity of mTORC1. We found that the phosphorylation levels of ribosomal protein S6 (p-S6) were significantly increased in *Trmt6⁻/⁻* LT-HSCs and LSK cells compared with those in *Trmt6⁺/⁺* LT-HSCs and LSK cells (Fig. 3e and S5a). *Trmt6⁻/⁻* HSCs also had other features of mTORC1 hyperactivation, such as increased cell size (Fig. S5b), increased ROS (Fig. S5c) and peak mitochondrial membrane potential changes (Fig. S5d). Although the apoptotic proportion of HSCs did not change significantly, the apoptotic rate of Lin⁺ cells was high (Fig. S5e). It is plausible that TRMT6 loss resulted in sustained mTORC1 activation, which was responsible for the phenotypic expansion and functional decline of HSCs.

m¹A at position 58 (m¹A58) in tRNA can be catalyzed by a methyltransferase complex, containing an RNA binding component TRMT6 and a catalytic component TRMT61A[34,44]. To further determine whether TRMT6 mediated methyltransferase complex exerted function through its enzymatic activity, we use quantitative mass spectrometry (LC-MS/MS) and RNA dot blot assay to analysis m¹A signals of big RNA ( > 200 nt RNA) and small RNA ( >50 nt and <200 nt RNA, mostly tRNA). We found that only global tRNA-m¹A58 modification levels in *Trmt6⁻/⁻* HSPCs were decreased compared with those of *Trmt6⁺/⁺* HSPCs (Figs. 3a, b).

tRNA-m¹A58 modification is required for efficient mRNA translation during initiation and elongation[45,59]. RT-PCR analysis of targets protein antibody found that the mRNA levels of the upstream targets (LKB1, AMPK, TSC1, GSK3β, REDD1, RHEB) of mTORC1 pathways[30] did not differ (Fig. 3f). Furthermore, FACS staining of target protein antibody further confirmed that the protein levels of the upstream targets (LKB1, AMPK, GSK3β, REDD1 and RHEB) of mTORC1 pathways did not differ (Fig. S5f), but TSC1 were significantly down-regulated in *Trmt6*-null LT-HSCs and LSK cells (Fig. 3g, S5g and S5h). To establish whether *Tsc1* mRNA, through its codon content, requires tRNA-m¹A58 modification during translation, we performed polyribosome RT-PCR experiments to quantify the ribosome occupancy of *Tsc1* mRNAs. Notably, we found a dramatic decrease in the accumulation of ribosomes on *Tsc1* mRNAs but not on LKB1, AMPK, GSK3β, REDD1, RHEB or control transcripts upon TRMT6 depletion, suggesting that tRNA-m¹A58 is more likely to regulate the translation elongation of *Tsc1* mRNA (Fig. 3h).

TSC1 deficiency leads to reduced reconstruction capacity of HSCs. To further explore the potential molecular mechanisms of TSC1 translational enhancement by tRNA-m¹A58 modification, we analyzed the tRNA-m¹A sequencing data of WT versus *Trmt6⁻/⁻* HSPCs to determine the magnitude of the reduction in m¹A58 modification levels on each tRNA after TRMT6 deletion (Figs. S6a–c). We found that the m¹A58 levels on most tRNAs were reduced to varying degrees after TRMT6 deletion, and the tRNAs decoding Alanine was among the most affected tRNAs (Fig. 3c). Alanine was also among the codons high frequently used by *Tsc1* mRNA (Fig. 3d and S6e). To further identify the transcripts whose translation were regulated by these m¹A58 elevated tRNAs, we performed ribosome profiling sequencing (Ribo-seq) analysis in TRMT6 depleted and control HSPCs. With Ribo-seq data, we found *Tsc1* gene has lower translation efficiency in *Trmt6⁻/⁻* HSPCs (Fig. S6f).

To determine whether TSC1 translation deficiency was due to mRNA translation initiation or elongation in *Trmt6⁻/⁻* HSPCs, we analyzed the m¹A58 levels and expression levels of initiator-methionine tRNA and found that neither was decreased in *Trmt6⁻/⁻* HSPCs (Fig. 3c, S6b and S6d), suggesting that tRNA-m¹A58 is more likely to regulate the translation elongation of *Tsc1* mRNA. Taken together, these results demonstrate that tRNA-m¹A58 is essential for efficient *Tsc1* mRNA translation in HSPCs.

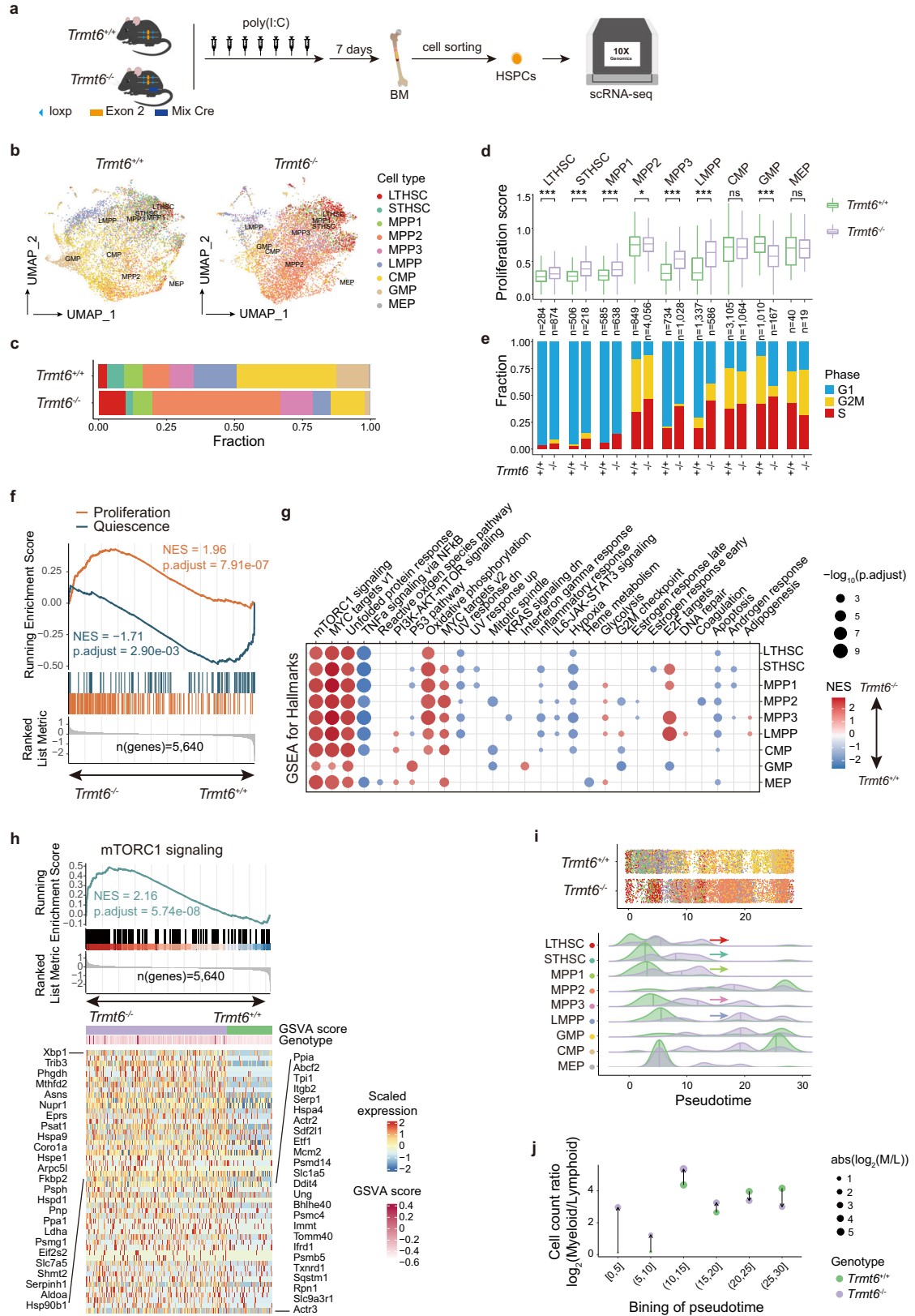

To further explore the potential molecular mechanisms of *Tsc1* translational enhancement by tRNA-m¹A58 modification, we reanalyzed the tRNA-m¹A sequencing data of WT versus *Trmt6⁻/⁻* HSPCs to determine the magnitude of the reduction in m¹A58 modification levels on each tRNA after TRMT6 deletion. Ala was among the codons most frequently used by *Tsc1* mRNA; thus, we remodeled *Tsc1* cDNA

such that the most abundant Alanine codons were replaced by their synonymous codons decoded by tRNAs least affected by TRMT6 deletion (Fig. S7a). Accordingly, the GCA/GCG (Ala) codons were replaced by GCC (Ala) (Fig. S7a). Expression of this mutant *Tsc1* via lentivirus in WT and *Trmt6⁻/⁻* HSPCs (Fig. S7b) was sufficient to rescue the defective expression of TSC1 (Fig. S7c) and single clone formation

**Fig. 2 | scRNA-seq revealed an activated mTORC1 signaling pathway in *Trmt6* deficient HSC populations. a** Experimental schematic for the generation of mice with bone marrow specific deletion of *Trmt6* and the obtaining of HSPCs for scRNA-seq. *Trmt6*[fl/+] mice were crossed with interferon-inducible transgenic Mx1-Cre mice to generate *Trmt6*[+/+] Mx1-Cre and *Trmt6*[fl/fl] Mx1-Cre mice. Mice were then treated with intraperitoneal injections of 300 ug/kg pIpC every other day seven times to delete the *Trmt6* alleles. ScRNA-seq was performed for FACS-purified HSPCs. **b** UMAP plots showing nine populations of *Trmt6*[+/+] HSPCs and *Trmt6*[-/-] HSPCs annotated with Nestorowa et al. data. Colors indicate cell types. **c** Histograms showing the compositions of nine populations. Colors indicate cell types. **d** Boxplot comparing the proliferation score of cells in each population between *Trmt6*[+/+] HSPCs and *Trmt6*[-/-] HSPCs. The standard boxplot notation was used (lower/upper hinges-first/third quartiles; whiskers extend from the hinges to the largest/smallest values no further than 1.5 x inter-quartile ranges; middle line-the median). The differences are tested by two-sided Wilcoxon rank sum test, *, $P < 0.05$; **, $P < 0.01$;

***, $P < 0.001$. **e** Histograms showing the proportions of nine populations in each cell cycle stages between *Trmt6*[+/+] and *Trmt6*[-/-]. **f** GSEA analyzes for genes affected in the LT-HSCs of *Trmt6*[-/-] versus *Trmt6*[+/+] control mice showing positive enrichment of proliferation signature (orange) and negative enrichment of quiescence signatures (blue). NES, normalized enrichment score. **g** Dot plot GSEA analyzes for genes affected in nine populations of *Trmt6*[-/-] versus *Trmt6*[+/+] to test enrichment for enrichment of hallmarks. **h** GSEA analyzes for genes affected in the LT-HSCs of *Trmt6*[-/-] versus *Trmt6*[+/+] showing positive enrichment of mTORC1 signaling. Heatmap shows the scaled expression of leading-edge subset of genes in mTORC1-signaling. **i** Jitter plots and density plots showing the distribution of pseudotime predicted by Monocle3 for nine populations. Colors indicate cell types. **j** Dot plot showing changes of the ratio between myeloid- and lymphoid-biased HSPCs after *Trmt6* deletion. The upward and downward arrows indicate higher or lower ration between myeloid- and lymphoid-biased HSPCs after *Trmt6* deletion, respectively. *P* values in d and f-h were BH-adjusted. Exact *P* values are provided as Source Data.

in *Trmt6*[-/-] HSCs in vitro (Fig. S7d), confirming that tRNA-m[1]A58 modification directly regulates *Tsc1* mRNA translation through codon decoding.

Furthermore, we knockdowned the expression of TRMT61A in HSCs, an essential m[1]A catalytic partner for TRMT6 in the m[1]A 'writer' complex (Fig. S8a). HSCs of sh-*Trmt61a* showed a similar phenotype to *Trmt6*-KO HSCs (Figs. S8b–g). Specifically, using the above methods, we found a similar self-renewal defect in HSCs of sh-*Trmt61* (Figs. S8c and d). Thus, our data demonstrate that *Trmt61a*-deficiency induced mTORC1 activation is due to the depletion of tRNA-m[1]A58 modification.

### Inhibition of mTOR pathway ameliorates aberrant proliferation and self-renewal defects of *Trmt6* knockout HSC

To examine the relationship between the enhanced mTOR signaling and dysregulated haematopoiesis in *Trmt6*[-/-] mice, we treated *Trmt6*[-/-] mice and age-matched control mice with rapamycin for 3 weeks (Fig. 4a). Importantly, treatment with rapamycin recovered the number of LT-HSCs, ST-HSCs, MPP2, MPP3 and MPP4 as well as B, T and myeloid cells of PB or spleen in *Trmt6*[-/-] mice compared with that in *Trmt6*[+/+] mice (Fig. 4b, c and S9a–c). Treatment with rapamycin recovered the cell cycle defects of LT-HSCs and LSK cells in *Trmt6*[-/-] mice compared with that in *Trmt6*[+/+] mice (Fig. 4d and S9d). Furthermore, there is a decrease in p-S6 levels in *Trmt6*[-/-] mouse after rapamycin treatment compared to vehicle, but no changes in *Trmt6*[+/+] mouse, suggesting that rapamycin treatment partially rescued the overactivation of mTOR signaling (Fig. 4e and S9e).

To test if TSC1 and TRMT6 or enzyme-dead TRMT6 over-expression can rescue the defect in vivo, *Trmt6*[-/-] and *Trmt6*[+/+] HSCs from primary mice were transduced with TSC1, TRMT6 or enzyme-dead TRMT6 (TRMT6[mut]) overexpression (OE) lentivirus, and trans-duced cells were transplanted into CD45.1 recipients (Figs. S10a and d). Chimerism in the bone marrow was evaluated at 16 weeks post-transplantation to read out long-term engraftment. We observed that OE of TSC1 can partially rescued the function of *Trmt6*-deficient HSCs (Figs. S10b and c). However, TRMT6[mut] OE did not improve *Trmt6*[-/-] HSCs' engraftment deficiency at the level of HSC and PB (Figs. S10e and f). The data suggested that TRMT6 play roles by the function of the m[1]A enzyme in HSCs, while TSC1 at least in part is responsible for TRMT6's role in maintaining reconstitution in HSCs

To investigate the effect of mTORC1 inhibition on the function-ality of *Trmt6*[-/-] HSCs in vivo, we conducted competitive transplanta-tion assays (Fig. 4f) and found that rapamycin treatment partially rescued the reconstitution ability of *Trmt6*[-/-] donor-derived HSCs in the PB and LT-HSC compartment of the recipients compared with vehicle treatment, indicating that mTORC1 inhibition also rescued the func-tion of *Trmt6*-deficient HSCs (Figs. 4g, h). Taken together, these data

suggest that overactive mTORC1 pathway partially causes dysregu-lated haematopoiesis in *Trmt6*[-/-] mice.

## Discussion

Maintaining the homeostasis and function of adult stem cells is essential to replenish somatic cells lost under physiological or patho-logical conditions[60,61]. How transcription and translation of HSCs are controlled is a central issue in haematopoietic homeostasis. Both the Wnt signaling pathway and the mTOR signaling pathway play impor-tant roles in the homeostasis and regeneration of HSC[3,8,17,24,57,62,63]. In our previous study, we found that the histone deacetylase *Sirt6* [8]and RNA m[6]A reader *Ythdf2*[20], maintains the function of HSCs by regulating the Wnt pathway. In contrast, in the present study, we found by single-cell RNA sequencing that the Wnt pathway was not affected in HSCs by *Trmt6* deletion, whereas the mTOR signaling pathway was significantly activated in *Trmt6*-deficient HSCs. Previous work has established that excessive activation of mTOR can severely impair the function of HSCs[24,27]. Thus, the level of mTORC1 activity may need to be precisely regulated at multiple levels in HSCs, and an increase or decrease in mTORC1 levels can lead to diminished haematopoietic stem cell function. Here, our study suggests that the TRMT6-tRNA m[1]A-TSC1-mTORC1 axis plays a key role in determining the homeostasis and regeneration of HSCs.

In order to protect the host from effects such as severe bleeding, HSCs must be able to respond quickly and renew and differentiate themselves[12,55,64]. This critical process is achieved by the timely synth-esis of large amounts of functional proteins[23,25,26]. Here, we demon-strate that tRNA-m[1]A58 methylation may act as an epigenetic translational control, constituting an important mechanism that enables the rapid synthesis of large amounts of functional proteins and drives activated HSCs into mitosis and rapid proliferation, and back to a resting state after activation. More specifically, through the use of genetic mouse models, high-throughput single-cell RNA sequencing, m[1]A-tRNA-seq, stem cell transplantation, we found that: (1) the tRNA-m[1]A58 'writer' protein (TRMT6) is one of the key genes for the main-tenance of resting state and self-renewal function in HSCs; (2) as tRNA-m[1]A58 promotes translation of TSC1 to control mTORC1 signaling in HSC at a lower level, it is possible that this is a strategy to better cope with the rapid translational demands of HSCs. As HSCs play a pivotal role in the haematopoietic system, changes in tRNA-m[1]A may provide a strategy to maintain the transcriptional state of HSCs once they are required to respond rapidly to different situations without the need to restart the complex and tightly controlled transcriptional process. From this point of view, tRNA m[1]A58 could be considered as a quality control system to protect the integrity of HSCs. This work also pro-vides mechanistic insights into the role of specific signaling pathways that control haematopoietic regeneration and will inform therapeutic strategies.

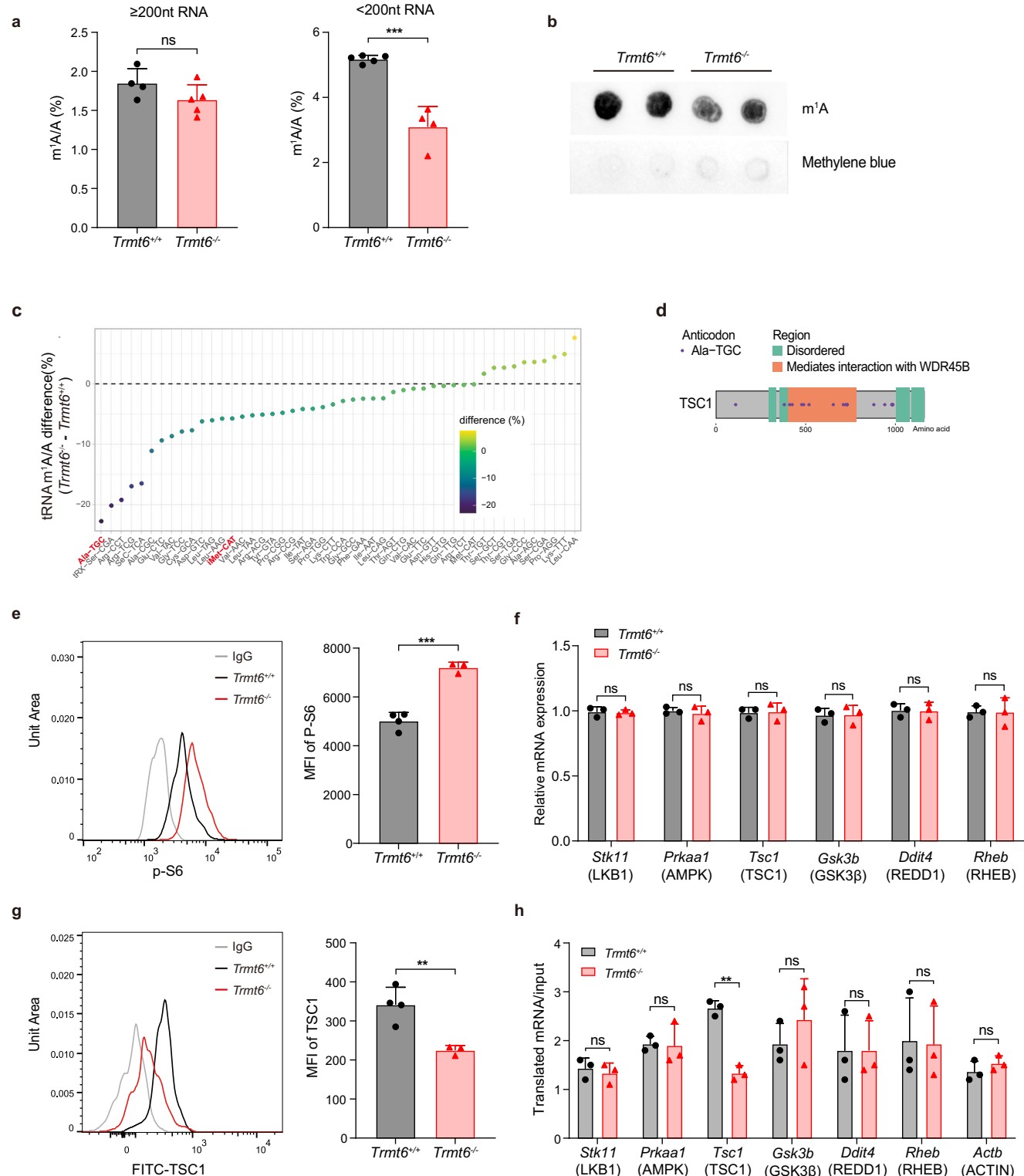

## Limitations of the study

According to published sequencing data, *Trmt6/Trmt61a* mRNA levels in HSCs is regulated by transplantation, radiotherapy, chemotherapy, and inflammatory stimuli (Fig. S1). Therefore, it is also important to figure out how and why *Trmt6* expression changes, which may disclose strategies to improve the function and lifespan of HSCs. mTORC1 hyperactivation in HSCs has been studied in mice with TSC1, PTEN and SZT2 deficiency. Recently, an important study reported that constitutive nutrient signaling to mTORC1 in the absence of SZT2 was detrimental to the repopulating capacity of the HSCs[65]. We found that

the phenotype of SZT2 deficiency in HSCs is similar with *Trmt6*-deficient HSCs. So How SZT2, TSC1, PTEN and TRMT6 synergistically regulate mTORC1 signaling in HSCs requires further investigation as well. Recently, He HQ et al. found TRMT6-TRMT61A over-expression regulate 3'-tiRNA-Leu-CAG → RIPK1 → RIPK3 → MLKL cascade to induce HSC aging through independent role of tRNA-m1A58 modification[46]. So, whether or not TRMT6-TRMT61A complex inhibitor prevents HSC aging is also an interesting topic for aging studies.

The focus of our current study is to elucidate the role of *Trmt6* on HSC homeostasis and self-renewal in mice. The role of TRMT6 in

**Fig. 3 | tRNA·m¹A modification is required for efficient *Tsc1* mRNA translation.**
**a** LC-MS/MS quantification of m¹A levels in the RNA (>200 nt) and tRNA (>50nt and <200 nt) purified from *Trmt6⁻ᐟ⁻* and *Trmt6⁺ᐟ⁺* HSPC, presented as percentage of unmodified A. *Trmt6⁺ᐟ⁺* HSPC RNA (>200 nt), *n* = 4 independent biosamples and tRNA, *n* = 5 independent biosamples. *Trmt6⁻ᐟ⁻* HSPC RNA (>200nt), *n* = 5 independent biosamples and tRNA, *n* = 4 independent biosamples. **b** Global m¹A levels were detected in tRNA (>50nt and <200 nt) purified from *Trmt6⁻ᐟ⁻* and *Trmt6⁺ᐟ⁺* HSPC using dot blot assay. Data were repeated five times. **c** The decrease in magnitude (the tRNA·m¹A58 level in *Trmt6⁻ᐟ⁻* HSPCs minus the level in *Trmt6⁺ᐟ⁺* HSPCs) of the tRNA·m¹A58 level in each tRNA after TRMT6 deletion. **d** The position of codon GCA (corresponding to tRNA-Ala-TGC) in mouse TSC1 protein. **e** Flow cytometry analysis of p-S6 levels in the LT-HSCs of *Trmt6⁻ᐟ⁻* and *Trmt6⁺ᐟ⁺* mice. Median fluorescence intensity of p-S6 in *Trmt6⁻ᐟ⁻* (*n* = 3 mice) and *Trmt6⁺ᐟ⁺* (*n* = 4 mice) LT-HSCs. **f** The relative mRNA expression levels of genes encoding LKB1, AMPK, TSC1, GSK3β, REDD1 and RHEB in *Trmt6⁻ᐟ⁻* LT-HSCs vs. *Trmt6⁺ᐟ⁺* LT-HSCs are shown. *n* = 4 independent biosamples. **g** Flow cytometry analysis of TSC1 levels in the LT-HSCs of *Trmt6⁻ᐟ⁻* and *Trmt6⁺ᐟ⁺* mice. Median fluorescence intensity of TSC1 in *Trmt6⁻ᐟ⁻* (*n* = 3 mice) and *Trmt6⁺ᐟ⁺* (*n* = 4 mice) LT-HSCs. **h** Ribosome occupancy of LKB1, AMPK, TSC1, GSK3β, REDD1, RHEB and control mRNAs was measured by RT-PCR as the relative expression ratio (RER) of polyribosome mRNAs to the input mRNAs after sucrose gradient fractionation of polyribosomes. Data represent the mean ± SD from three independent experiments. *, *P* < 0.05; **, *P* < 0.01; ***, *P* < 0.001. For all the above statistics, *P* values were obtained using unpaired parametric two-tailed t-test. Exact *P* values are provided as Source Data.

human HSC function and senescence is also an interesting topic for future studies.

# Methods

## Resource availability

**Lead contact.** Further information and requests for resources and reagents should be directed to and will be fulfilled by the lead contact, Hu Wang (wanghu19860315@163.com).

**Materials availability.** All biological materials used in this study are available from the lead contact upon request or from commercial sources.

**Mice.** *Trmt6*-floxed mice were generated at Cyagen Biosciences Inc. *Trmt6*ᶠˡ⁄ᶠˡ mice were mated to Mx1-Cre transgenic mice to generate *Trmt6*⁺ᐟᶠˡ Mx1-Cre and *Trmt6*ᶠˡ⁄ᶠˡ Mx1-Cre mice. The *Trmt6*ᶠˡ⁄ᶠˡ mice genotyping forward primer: 5′-GCTCACGAGATGATGGTGGGGA-3′, reverse primer: 5′-GTATACACTGGAAGCTCAGGGCTAATG-3′. All of these strains were maintained on a C57BL/6 background. The recipient mice used in the competitive transplantation assays were either B6.SJL-PtprcaPep3b/Boy (CD45.1) mice. Both male and female mice were utilized in the study. The Animal Care and Ethics Committee at Hangzhou Normal University approved all animal experiments in this study. All mice were maintained in a pathogen-free environment and fed a standard diet.

**Flow cytometry.** Prepared samples were analyzed on an LSRFortessa™ cell analyzer (BD Biosciences) or sorted on an MoFlo Astrios EQ cell sorter (Beckman colter). For whole bone marrow cells, HSPC were stained with lingure cocktail (CD11B, CD4, CD8, B220, Gr1, Ter119), c-Kit, Sca1, Flt3, CD150, CD48, CD34, CD16/32 and IL-7R and then analyzed by flow cytometry for HSPC cell types and proportions. Bone marrow cells were also incubated with lingerie cocktail (CD11B, CD4, CD8, B220, Gr1, Ter119), c-Kit, Sca1, CD150, CD48 and then fixed on rupture membranes and analyzed for HSC cell cycle using Ki67 and DAPI; or stained with antibodies against p-S6 and TSC1 to detect changes in protein levels. For peripheral blood and spleen cells, CD3, CD11b and B220 were used to determine the proportion of B cells, T cells and myeloid cells.

**Isolation of Murine HSC.** Donor mice at 8 weeks of age were euthanized by cervical dislocation, and bone marrow (BM) cells were isolated by crushing the bones from hind legs, forelegs, pelvis, spine and sternum. BM cells were firstly enriched for HSCs by anti-CD117 conjugated magnetic beads (Miltenyi) and followed by FACS sorting.

**Transplantation Assay.** Competitive transplantation: 1 × 10⁶ cells from donor mice (CD45.2) were mixed with 1 × 10⁶ BM MNCs from age-matched WT mice (competitor, CD45.1) and injected into lethally irradiated recipient mice (CD45.1).

**Rapamycin treatment.** Rapamycin was dissolved in Ethanol (20 mg/mL) and further diluted with corn oil. Mice were treated with 4 mg/kg rapamycin intraperitoneally every other day.

**RNA Purification and RT-PCR.** Total RNA was extracted from freshly isolated cells using a RNeasy kit (Qiagen). Complementary DNA was then synthesized from the RNA using a HiScript II One Step RT-PCR Kit (Vazyme Biotech Co.Ltd) for First-Strand cDNA Synthesis according to the manufacturer's instructions. The quantitative RT-PCR analysis was conducted using a CFX96 Real-Time System (Bio-Rad). The relative expression levels of the genes of interest were calculated using the relative delta-delta-Ct method. The expression of β-actin was used as the internal control. The primer sequences used for real-time PCR were as follows:

*Gsk3b* forward primer: 5′-TGGCAGCAAGGTAACCACAG-3′ and reverse primer: 5′-CGGTTCTTAAATCGCTTGTCCTG-3′

*Ddit4* forward primer: 5′-CAAGGCAAGAGCTGCCATAG-3′ and reverse primer: 5′-CCGGTACTTAGCGTCAGGG-3′

*Tsc1* forward primer: 5′-ATGGCCCAGTTAGCCAACATT-3′ and reverse primer: 5′-CAGAATTGAGGGACTCCTTGAAG-3′

*Stk11* forward primer: 5′-TTGGGCCTTTTCTCCGAGG-3′ and reverse primer: 5′-CAGGTCCCCCATCAGGTACT-3′

*Rheb* forward primer: 5′-AAGTCCCGGAAGATCGCCA-3′ and reverse primer: 5′-GGTTGGATCGTAGGAATCAACAA-3′

*Prkaa1* forward primer: 5′-GTCAAAGCCGACCCAATGATA-3′ and reverse primer: 5′-CGTACACGCAAATAATAGGGGTT-3′

**Quantification of m¹A levels by LC-MS/MS.** Adding buffer, S1 nuclease, Alkaline Phosphatase and Phosphodiesterase I into 1 μg RNA, then the mixture was incubated at 37°C. After the RNA was digested into nucleosides completely, the mixture was extracted with chloroform. The resulting aqueous layer was collected for analysis with LC-ESI-MS/MS. The sample extracts were analyzed using an UPLC-ESI-MS/MS system (UPLC' ExionLC™ AD' https: //sciex.com.cn/ ; MS' Applied Biosystems 6500 Triple Quadrupole, https://sciex.com. cn/). The analytical conditions were as follows, LC: column, Waters ACQUITY UPLC HSS T3 C18 (1.8 μm, 2.1 mm × 100 mm); solvent system, water (2 mM NH4HCO3): methanol (2 mM NH4HCO3); gradient program, 95:5 V/V at 0 min, 95:5 V/V at 1 min, 5:95 V/V at 9 min, 5:95 V/V at 11 min, 95:5 V/V at 11.1 min, 95:5 V/V at 14 min; flow rate, 0.30 mL/min; temperature, 40 °C; injection volume: 10 μL. The effluent was alternatively connected to an ESI-triple quadrupole-linear ion trap (QTRAP)-MS. Linear ion trap (LIT) and triple quadrupole (QQQ) scans were acquired on a triple quadrupole-linear ion trap mass spectrometer (QTRAP), QTRAP® 6500 + LC-MS/MS System, equipped with an ESI Turbo Ion-Spray interface, operating in positive ion mode and controlled by Analyst 1.6.3 software (Sciex). The ESI source operation parameters were as follows: ion source, ESI + ; source temperature 550 °C; ion spray voltage (IS) 5500 V; curtain gas (CUR) was set at 35 psi, respectively. RNA modifications were analyzed using scheduled multiple reaction monitoring (MRM). Data

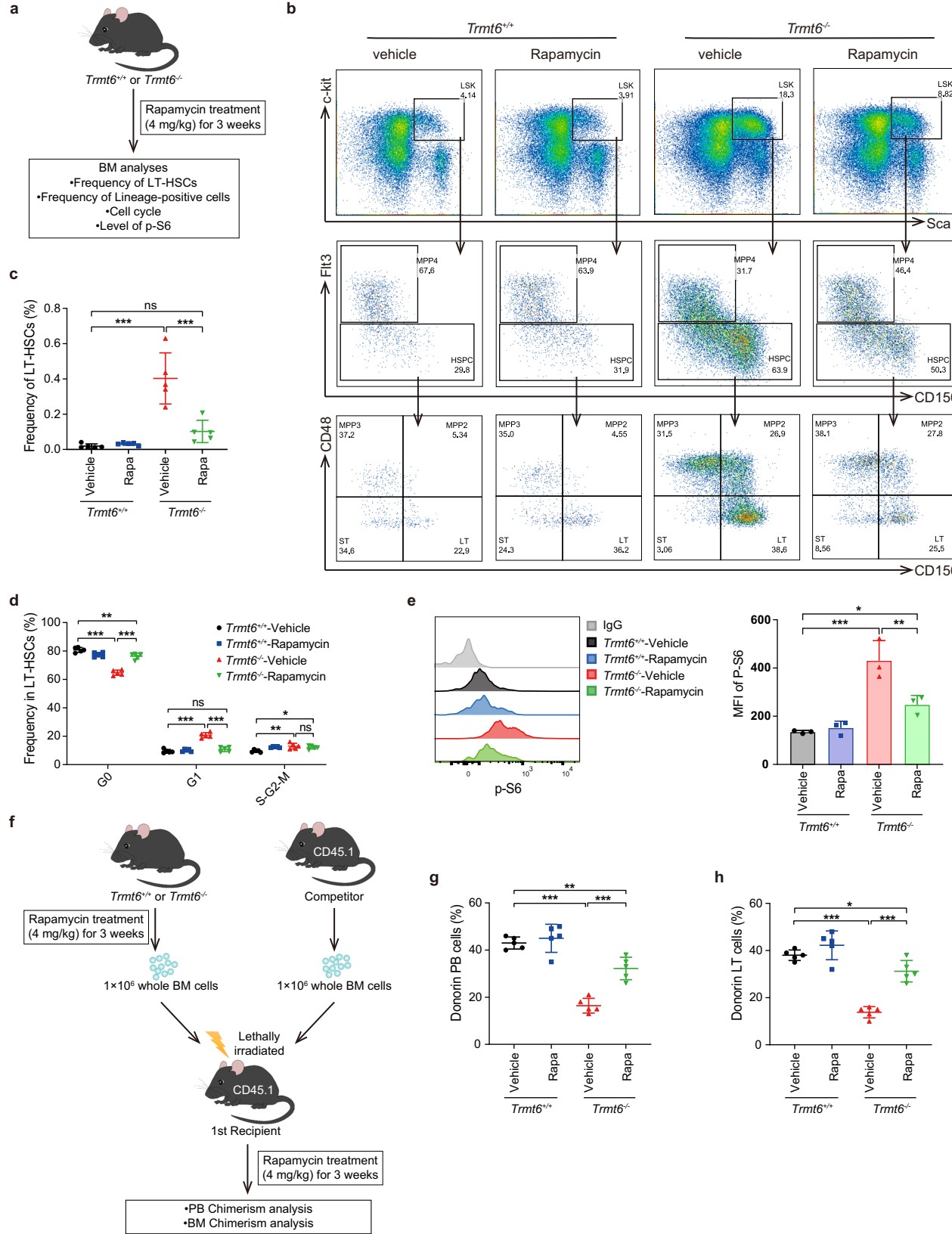

acquisitions were performed using Analyst 1.6.3 software (Sciex). Multiquant 3.0.3 software (Sciex) was used to quantify all metabolites. Mass spectrometer parameters including the declustering potentials (DP) and collision energies (CE) for individual MRM transitions were done with further DP and CE optimization. A specific set of MRM transitions were monitored for each period according to the metabolites eluted within this period.

**Dot blot assay.** Total RNA was extracted by TRIZOL (Invitrogen). Purified tRNA was quantified and diluted in 10 mM Tris-EDTA buffer.

**Fig. 4 | Inhibition of mTOR pathway ameliorates aberrant proliferation and self-renewal defects of *Trmt6* knockout HSC. a** The experimental design for treatment with an mTOR inhibitor rapamycin in *Trmt6⁻/⁻* and *Trmt6⁺/⁺* mice. **b** FACS analysis of LT-HSCs, ST-HSCs and MPPs in *Trmt6⁻/⁻* and *Trmt6⁺/⁺* BM cells after mTOR inhibitor rapamycin treatment in vivo. Representative FACS profiles are shown. *n* = 5 mice per genotype. **c** Frequency of LT-HSCs in BM cells are shown after mTOR inhibitor rapamycin treatment in vivo. *n* = 5 mice per genotype. **d** Experimental schematic for serial competitive transplantation with *Trmt6⁻/⁻* and *Trmt6⁺/⁺* BM cells after mTOR inhibitor rapamycin treatment in vivo. **e** Flow cytometry analysis of p-S6 levels in the LT-HSCs of *Trmt6⁻/⁻* and *Trmt6⁺/⁺* mice after mTOR inhibitor rapamycin treatment in vivo. Median fluorescence intensity of p-S6 in *Trmt6⁻/⁻* and *Trmt6⁺/⁺* LT-HSCs after mTOR inhibitor rapamycin treatment in vivo. *n* = 3 mice per

genotype. **f** The experimental design for transplantation using whole bone marrow cells treated with rapamycin. Whole bone marrow cells isolated from *Trmt6⁻/⁻* and *Trmt6⁺/⁺* mice that had been treated with rapamycin for 3 weeks were transplanted into lethally irradiated recipient mice with competitor cells. Recipient mice were continuously treated with rapamycin after transplantation (results in Fig. 4g–h). **g** After transplantation, the frequency of donor-derived cells in peripheral blood were analyzed. *n* = 5 mice per genotype. **h** After transplantation, the frequency of donor-derived cells in LT-HSCs were analyzed. n = 5 mice per genotype. Data represent the mean ± SD from three independent experiments. ns, *P* value ≥ 0.05; *, *P* < 0.05; **, *P* < 0.01; ***, *P* < 0.001. For all the above statistics, *P* values were obtained using one-way ANOVA followed Dunnett's multiple comparisons test. Exact *P* values are provided as Source Data.

100–200 ng RNA were denatured at 75 °C for 5 min and loaded to positively charged nylon membrane. The membrane was UV-crosslinked with 2400 J/cm2 twice and then probed with anti-m¹A antibody, followed with HRP-conjugated secondary antibody staining and ECL detection.

**Single cell RNA-seq and analysis.** For sorted LSK cells from *Trmt6⁺/⁺* or *Trmt6⁻/⁻* BM, single cells were captured in droplet emulsions using a Chromium Controller (10X Genomics), and scRNA-seq libraries were constructed by following the 10X Genomics protocol using Chromium Single Cell 3' Reagent Kits v3. Raw sequencing data were converted to fastq format using the command 'cellranger mkfastq' (10x Genomics, v.6.0.0). scRNA-seq reads were aligned to the GRCm38 (mm10) reference genome and quantified using 'cellranger count' (10x Genomics, v.6.0.0). Count data was further processed using the 'Seurat' R package (v. 4.1.1). Genes detected in less than 3 cells were excluded, and cells were required to have >200 and not too many genes ( <4000 for *Trmt6* WT and <4500 for *Trmt6* cKO mice), not too many mapped reads mapping to mitochondrial genes ( <10% for *Trmt6* WT and <5% for *Trmt6* cKO mice), less than 5% of mapped reads mapping to hemoglobin genes. 'DoubletFinder' R package (v.2.0.2) was used to find and remove doublets. After the above filtering, 8450 and 8650 cells were obtained for *Trmt6* WT and *Trmt6* KO samples, respectively. Data were normalized by Seurat's 'NormalizeData' function. For cell type annotation, the Sonia Nestorowa et al. data was downloaded and Spearman's association test was performed between our single cell population and stem/progenitor cell population in Sonia Nestorowa et al. article with SingleR R package, the SingleR score was used as an indicator of the similarity between our single cell data and the Sonia Nestorowa et al. data. Gene expression signatures for each cell type were calculated using the Seurat's 'FindAllMarkers' function by looking at genes detected in a minimum of 10% of cells and with higher expression level (log2FC ≥ 0.25) in one cell type versus in other cells. Cells were assigned to either 'G2/M' phase or 'S' phase using the CellCycleScoring function of the Seurat package according to previously defined cell cycle genes specific to either G2/M phase or S phase. Cells expressing none of these genes were assigned to 'G1' phase. Proliferation scores were calculated by averaging expression of known proliferation-related genes using the Average Expression function in Seurat. The proliferation-related genes are *Aurka, Bub1, Ccnb1, Ccnd1, Ccne1, Dek, Fen1, Foxm1, H2afz, Hmgb2, Mcm2, Mcm3, Mcm4, Mcm5, Mcm6, Mki67, Mybl2, Pcna, Plk1, Top2a, Tyms, Zwint.* The difference of proliferation score between *Trmt6⁻/⁻* and *Trmt6⁺/⁺* cells of each type was tested using two-sided Wilcoxon rank sum test.

Data of two samples were integrated using 3000 features after normalization individually by Seurat's 'SCTransform' function with parameter 'vars.to.regress' set as percentage of mitochondrial read count and the difference between cell cycle scores of 'S' and 'G2/M' phases. Principal component analysis (PCA) was performed, nearest neighbors were computed on the basis of the first 30 dimensions. Uniform manifold approximation and projection (UMAP) analysis was

performed also on the basis of the first 30 dimensions using 'RunUMAP' function in Seurat with default parameter settings. To identify genes whose expression are affected by *Trmt6* depletion for each cell type, differentially expressed genes between KO and WT cells of each type were identified by Seurat's 'FindMarkers' function. Gene Set Enrichment Analysis (GSEA) were performed using clusterProfiler's 'GSEA' function on genes ranked by log2-transformed fold change of gene expression in *Trmt6*-KO cells. The mouse-ortholog hallmark gene sets downloaded from MSigDB (https://www.gsea-msigdb.org/gsea/msigdb) and signature genes for proliferation and quiescence were obtained from published data. The GSVA score of hallmark gene set 'mTORC1 signaling' were calculated using R package GSVA's 'gsva' function. To analyze intercellular communication, CellphoneDB (v4.0.0) was used to identify significant ligand-receptor pairs with *P* values < 0.05 for receptors and ligands expressed by more than 15% of cells of each cell type.

**tRNA-Seq library construction.** The tRNASeq libraries were prepared using the mim-tRNAseq workflow[66,67]. Briefly, the RNA samples were decylated by in Tris-Cl (PH9), followed by dephosphorylation with T4 PNK (NEB, M0201S) and ethanol precipitation. The RNA samples were separated on 10% TBE-Urea PAGE gel. RNA of 60–100 nt in length was recovered by gel excision and elution from gel slices, followed by ethanol precipitation. The gel-purified tRNA was then ligated to the preadenylated barcoded 3'-aptamer using T4 RNA Ligase 2, truncated KQ (NEB, M0373L). The mix was incubated for 3 h at 25 °C and the ligation products were purified by size selection on a 10% TBE-Urea PAGE gel. The aptamer-conjugated tRNA (100 ng) was ligated to 1 µl of a 1.25 µM RT primer (5'-pRNAGATCGGAAGAGAGCGTCGTGTAGG GAAAGAG /iSp18/GTGACTGGAGTTCAGACGTGTGTGCTC-3'). Anneal at 82 °C for 2 min, then incubate at 25 °C for 5 min. Incubate with 500 nM TGIRT in 50 mM Tris-HCl pH 8.3, 75 mM KCl, 3 mM MgCl2, 5 mM DTT, and 20 U Superase In solution at 22 °C for 30 min; reverse transcription was performed with the addition of 1.25 mM dNTPs, and the incubation was performed at 42 °C for 16 h. After reverse transcription, NaOH was added to a final concentration of 0.1 M and the RNA was hydrolyzed by incubating the samples for 5 min at 90 °C. Complementary DNA products and unextended primers were separated on a 10% TBE-Urea PAGE gel. Gel-purified and ethanol-precipitated cDNA was incubated with CircLigase ssDNA ligase (Lucigen) in 1×reaction buffer, 1 mM ATP, 50 mM MgCl2 and 1 M betaine for 3 hours at 60 °C. The enzyme was inactivated at 80 °C for 10 min, and one-fifth of the cyclized cDNA was used directly for library construction PCR with a common forward(5'-AATGATACGGCGACCACCGA-GATCTACACTCTTTCCCTACACGACGCT∗C-3') and unique indexed reverse primers (5'-CAAGCAGAAGACGGCA TACGAGATNNNNNN GTGACTGGAGTTCAGACGTGT∗G-3'; NNNNNN, the reverse complement of an Illumina index sequence) with KAPA HiFi DNA polymerase (Roche) in 1×GC buffer with an initial denaturation at 95 °C for 3 min, followed by five cycles of 98 °C for 20 s, 62 °C for 30 s and 72 °C for 30 s at a ramp rate of 3 °C/ sec. The PCR product was purified with a

DNA Clean and Concentrator 5 kit (Zymo Research) and analyzed with a Qubit dsDNA HS kit (Thermo Fisher Scientific, Q32851). The PCR products were purified with a DNA Clean and Concentrator 5 kit (Zymo Research), quantified with a Qubit dsDNA HS kit (Thermo Fisher Scientific, Q32851), and sequenced on the Illumina Novaseq 6000 platform.

**mim-tRNA-seq data analysis.** Read demultiplexing and adapter trimming performed with cutadapt (v2.5), where bases were quality trimmed from 5′ and 3′ ends using a phred score cutoff of 30. Only trimmed reads were retained and reads shorter than 10 bases were discarded. An additional round of trimming was performed to remove 5′-RN nucleotides introduced by circularization from the RT primer. Reads were processed with the mim-tRNAseq computational pipeline using the common parameters --species Mmus --cluster-id 0.97 --threads 15 --min-cov 0.0005 --max-mismatches 0.075 --local-modomics --max-multi 4 --remap --remap-mismatches 0.05.

**Ribo-seq sequencing.** Cell lysates were treated with non-specific nucleic acid nuclease RNase I. Monomers were separated by size exclusion chromatography using a MicroSpinS-400HR column. Prior to PAGE purification of relatively short (20-38 nt) RPFs, RRNA samples were treated with an RRNA Decontamination Kit (Qiagen) to remove as much RRNA contamination as possible. After library construction (Multiplex Small RNA Library Prep Set for Illumina (Set1), NEB), the concentration of the libraries was measured with a Qubit® 2.0 fluorometer and adjusted to 1 ng/μL. The insertion of the obtained libraries was detected using an Agilent 2100Bioanalyzer, and finally the DNA libraries were detected again using a qPCR to detect the exact concentration of the DNA library again. After library preparation and pooling of different samples, the samples were subjected to luciferase sequencing.

**Bulk RNA-seq data analysis.** Raw reads downloaded from SRA were filtered using fastp (v0.23.2, default parameters) and mapped to mm10 using HISAT2 (v2.2.1), with parameters "--rna-strandness RF" for stranded RNA-seq data. Read counts per gene were calculated using featureCounts (v2.0.3). Differential expression analyses were performed using DESeq2 (v1.34.0).

**Codon-switch assay.** The coding DNA sequences of *WT-Tsc1* and *Mut-Tsc1* were constructed into a pLV-SFFV-IRES-eGFP plasmid. Lentivirus was produced in 293 T cells using the standard protocols. Transfection was performed using X-tremeGENE HP DNA Transfection Reagent (Roche). Purified WT and *Trmt6⁻/⁻* HSCs were activated in a 96-well plate for 12 h and then infected with the packaged lentivirus by spinning at 1000 g for 90 min. Cells were collected for immunoblot analysis and single clone formation analysis 48 h after infection at 37 °C.

**Quantification and statistical analysis.** All data represent mean ± SD. The unpaired parametric Two-tailed t-test and one-way ANOVA were utilized to determine statistical significance; $P$ values less than 0.05 were considered significant. The survival curve was analyzed using a log-rank (Mantel-Cox) test.

### Reporting summary
Further information on research design is available in the Nature Portfolio Reporting Summary linked to this article.

## Data availability
The scRNA-seq and mim-tRNAseq data generated in this study have been deposited in the GEO database under accession code GSE252914. Other single cell or bulk RNA-seq data of hematopoietic cells were downloaded from GEO database under accession code GSE116530, GSE151799, GSE143655, GSE59114 and SRA database under accession PRJNA717283. Source data are provided in this paper.

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

## Acknowledgements

This work was supported by the National Key R&D Program of China (2021YFA1102800) and by National Natural Science Foundation of China (92249304, 82301746). This work was supported by the Science Foundation for Distinguished Young Scholars of Guangdong Province

(2019B151502008) to Hu Wang. This work was supported by Interdisciplinary Research Project of Hangzhou Normal University (2024JCXK05) and the Hangzhou Youth Innovation Team Project (TD2023020).

## Author contributions

H.W. initiated the study and developed the concept of the paper. H.N.Z, A.W.W., M.W.W., L.Q.H., and H.W. conceived the project, designed experiments, and wrote the manuscript. H.N.Z and A.W.W. performed and analyzed all experiments. A.W.W. performed computational analysis.

## Competing interests

The authors declare no competing interests.
