## [Peer Review File · Nature Communications]

tRNA m1A modification regulate HSC maintenance and self-renewal via mTORC1 signalingREVIEWER COMMENTS

Reviewer #1 (Remarks to the Author):

The authors investigated the role of TRMT6 in HSC, and found that deletion of TRMT6 in hematopoietic cells via Mx1-Cre resulted in defective hematopoiesis and reduced repopulating potential of HPSC. Mechanistic studies revealed hyperactivation of mTORC1 as the cause of HSC defects. The experiments were well-performed and most conclusions are appropriate. However, a key weakness of this study is the lack of mechanism of TRMT6 regulation of TSC1.

1. Line 30, in the abstract, it was mentioned that “Trmt6-null mice became moribund due to haematopoietic failure with pancytopenia in the blood and bone marrow 4 to 8 weeks after Trmt6 deletion”. The authors did not provide data about these conclusions, such as survival curve of KO mice. And, “pancytopenia” was not evident from Fig. S1H-S1L, it seems that KO mice have selective reduction of B cells.
2. Line 118, “Those results suggested a bias in the lineage output towards to myeloid lineages” is not accurate. In Fig. S1G, T cells in PB from KO mice were increased. Thus, there was a selective reduction of B cells, but not T cells, which does not support myeloid bias. Absolute cell number of T, B and myeloid cells should be provided.
3. Whether the phenotypes were dependent on enzyme activity of TRMT6 was not addressed. Rescue experiments with WT TRMT6 and enzyme activity defective mutant of TRMT6 should be performed to answer the question. It is possible that TRMT6 might regulate HSC independent of RNA modification.
4. Line 228, “TSC1 were significantly upregulated in Trmt6-null LT-HSCs and LSK cells (Figures 3E and S4B)”. TSC1 was downregulated, not upregulated. Also, the level of TSC1 should be further confirmed by WB. Partial reduction of TSC1 is unlikely to cause a dramatic increase of pS6, since TSC1^{+/-} mice do not have obvious increase of pS6 and defects in hematopoiesis. Does TRMT6-KO HSC have other features of mTORC1 hyperactivation, such as increase cell size? Increase ROS? Whether overexpression of TSC1 could rescue the KO phenotypes? These are weak points of this study.
5. How TRMT6 specifically regulates translation of Tsc1 but not other mRNA should be explored or discussed.
6. The phenotype of TRMT6-KO is most close to TSC1/SZT2 KO mice reported recently, which should be discussed.

Reviewer #2 (Remarks to the Author):

The manuscript “tRNA m1A modification regulate HSC maintenance and self-renewal via mTORC1 signaling” aims at examining how the m1A modification impacts hematopoiesis. The authors find TRMT6 (the binding component of the m1A methyltransferase complex) is upregulated following transplantation, and treatment of HSCs with ionizing radiation, LPS etc. To begin to understand the role of TRMT6-mediated m1A modification in hematopoiesis, the authors generated a *Trmt6* conditional knockout mouse, and determined that the loss of *Trmt6* impacts HSC quiescence and self-renewal. The authors also show that the mTORC1 signaling pathway, which is essential for HSC maintenance and self-renewal, is upregulated in HSC-enriched cell populations following *Trmt6* deletion. The authors then try to link loss of *Trmt6*, with reduced m1A modification in tRNAs, and suggest that this results in less efficiently translated TSC1 mRNA (a negative regulator of mTOR).

My major concern is with the lack of detail in the proposed mechanism. The rescue experiments in Figure 4 show that the mTORC1 pathway, can ameliorate the defects in self-renewal and proliferation of HSCs, following loss of *Trmt6*. However, the mechanism is not at all convincing without the addition of m1A-tRNA-seq analysis, RiboTag, and differential protein analysis. Essentially the authors claim that loss of *Trmt6* results in less m1A tRNA modification which in turn specifically results in reduced translation of TSC1, and in turn aberrant mTOR signaling and deficiencies in homeostasis and regeneration of HSCs. Can the authors specifically show reduction of m1A modification on specific tRNAs? The authors have not mentioned how much RNA they isolate. But as a little as 200 ng of small RNAs could be used for m1A-seq. The authors should check how tRNA expression levels change following *Trmt6* deletion. Authors should specifically determine how the tRNA pool changes in terms of expression and/or modification levels following *Trmt6* deletion. The authors will need to check whether loss of m1A modification is related to translation initiation or elongation. It could be an initiation effect. The m1A modification is key to stability of the initiation tRNA. The authors may need to differentiate initiation from elongation effects. Following *Trmt6* deletion, the authors should check for the enrichment of hypo-m1A modified tRNA codons in TSC1 compared to other genes? If this is the case, it would be nice to see a codon switch experiment to confirm this. However, considering m1A is found on many tRNA species, the authors should determine the whole set of mRNAs whose translation efficiency is affected. To this end, authors should cross a RiboTag mouse with their Cre mouse and interrogate ribosome-associated transcriptomes in their system at a global level. Following that, the authors should confirm differential protein expression. Authors should at least show differential protein expression of TSC1.

In addition to TRMT6 what other genes involved in translation change upon transplantation, LPS etc? Can the authors compare changes of TRMT6 with other genes? A figure showing the most significant changes in genes including transcriptional factors and translation components would be good. Along those lines, did the authors look at TRMT6^{1A} expression changes upon transplantation,

radiotherapy, chemotherapy, and inflammatory stimuli? Do other 'writers', 'readers' and 'erasers' change expression?

Considering m6A has been shown to be preferentially required for adult HSC differentiation in vivo and in vitro, the authors should check that TSC1 is not marked by m6A, seeing that m6A is known to affect translation.

Minor points:

Figure 1: B) Mix Cre should be Mx1-Cre

Page 12 - Line 229 do the authors mean upregulated in Trmt6-null or down regulated?

Page 13 – I would be more careful with the wording used in section “Inhibition of mTOR pathway ameliorates aberrant proliferation and self-renewal defects of Trmt6 knockout HSC”. For example, Line 247 – 249. There is a decrease in p-S6 levels in Trm6 KO mouse after rapamycin treatment compared to vehicle (it is not a decrease compared to Trmt6 WT mouse). Or just say levels of p-S6 are rescued following rapamycin treatment in Trmt6 knockout mouse.

Authors should include amounts of RNA used in the methods section.

Reviewer #3 (Remarks to the Author):

In this study, Zuo H et al studied the roles and mechanism of Trmt6 in regulating HSC maintenance and self-renewal. Using a conditional Trmt6 KO mouse model, they found that Trmt6 KO resulted in the transition of quiescent HSCs into the cell cycle, increased HSPC frequencies and decreased reconstitution potential. Mechanistically, Trmt6 KO led to translational repression of Tsc1, which activated mTORC1 signaling pathway and led to HSC exhaustion. Finally, they found rapamycin ameliorates aberrant proliferation and self-renewal defects of Trmt6 KO HSCs. Overall, the role and regulatory mechanisms of Trmt6 in adult HSC maintenance are still largely unknown. The study is carefully designed, and the manuscript is well written. However, this study is rather preliminary with a lot of phenotypic data, but lacks comprehensive mechanistic investigation. Several issues need to be addressed.

Major issues:

1. The authors claim the importance of tRNA m1A modification in the regulation of HSC maintenance and self-renewal. Given Trmt6 and Trmt61a form m1A methyltransferase complex, it would strengthen their conclusion by doing these analyses. (1) To examine the expression of Trmt6 gene across multiple hematopoietic cell populations utilizing publicly available single-cell transcriptomic data. (2) To check whether Trmt61a KO phenocopies the HSC defects in Trmt6 KO mice. (3) To demonstrate whether the tRNA m1A-TSC1-mTORC1 axis is also responsible for Trmt61a KO HSCs.

2. In Fig. S1, the authors showed that Trmt6 was dysregulated in HSC upon different treatment or during aging. However, they only examined the effect of Trmt6 in steady hematopoiesis. To further consolidate their data, they should check the effect of Trmt6 on HSC number and functions under stress conditions, such as 5-Fu, irradiation and aging.

3. In Fig. 1, the authors performed single-cell RNA sequencing (scRNA-seq) on sorted LSK cells isolated from bone marrow (BM) of young adult Trmt6^{+/+} and Trmt6^{-/-} mice. However, the UMAP clustering analysis in Fig. 1C is not convincing because most of the HSPC clusters are hard to distinguish, particularly for LT-HSC, ST-HSC and MPPs. The authors should check the expression of known markers for these populations, such as Sca1, c-Kit, CD34, FLK2, CD150, CD48, CD127, etc.

4. In Fig. 2, the authors conducted competitive transplantation experiments by transplanting the same amount of total bone marrow cells and found that Trmt6 deletion produces a cell-autonomous functional defect in HSCs. Since the frequency of LT-HSCs is increased in WT and Trmt6 KO mice, it would be better to sort the LT-HSC and transplant with equal number of LT-HSC.

5. In the whole study, the authors evaluated the effect of Trmt6 on HSCs by use of Mx1-cre; Trmt6 floxed mice. They gave the mice with 6 times of poly I:C injections and performed the experiments and scRNA-seq only 1 week after injection. In the HSC field, many groups usually examined the HSC phenotype 3-4 weeks after 6 times of poly I:C injections so that the genetic deletion would be more complete. Considering that Trmt6 was not completely deleted in LSK cells (Fig. S1G), they should check the HSC phenotype after a longer time (over 4 weeks after poly I:C injection) to evaluate whether the effect was stronger.

6. In Fig. 2B-D, the authors showed that the frequencies of HSPCs and committed progenitors were all increased. Besides, they showed the frequency of B cells was reduced whereas the frequency of

myeloid cells was augmented (Fig. S1J-K). However, the bone marrow cellularity was declined in Fig. S1L. Could the authors explain why the total bone marrow cell number was dramatically decreased in the context of HSPC expansion?

7. Given that Trmt6 KO led to activation of mTORC1 signaling pathway and HSC exhaustion, whether apoptosis rate and ROS levels are also increased in the Trmt6 KO HSCs? And whether mitochondrial metabolism is also altered in the Trmt6 KO HSCs?

8. In Fig.3, they found that the protein levels of the upstream targets (LKB1, AMPK, GSK3 β , REDD1 and RHEB) of mTORC1 pathways did not differ (Figure S4C), but TSC1 were significantly upregulated in Trmt6-null LT-HSCs and LSK cells (Figures 3E and S4B). Besides, they performed polyribosome RT-PCR experiments to quantify the ribosome occupancy of Tsc1 mRNAs, and found a dramatic decrease in the accumulation of ribosomes on Tsc1 mRNAs but not on LKB1, AMPK, GSK3 β , REDD1, RHEB or control transcripts upon TRMT6 depletion. This is quite interesting, since it looks like Trmt6 specifically regulated the translation of Tsc1. How is the specificity executed by TRMT6? The authors should provide more mechanistic investigation into these interesting findings.

Minor issues:

1. Beyond the flow cytometry, Hemogram analysis should be performed in the conditional Trmt6 KO mice to evaluate the detailed information of all the blood cells.

2. Fig.2A and Fig.2D were not mentioned in the result session.

3. In the abstract, the authors mentioned that “Trmt6-null mice became moribund due to hematopoietic failure with pancytopenia in the blood and bone marrow 4 to 8 weeks after Trmt6 deletion.” However, there was no data on this conclusion in the manuscript.

4. The abstract should be re-written since a lot of useful information was not mentioned.

5. The authors showed the scRNA-seq data in the Fig.1. It might be better to present the phenotypic and transplantation results (Fig.2) in the Fig.1, and put the scRNA-seq data in the Fig.2. In this case, it would be easier to understand why they needed to perform scRNA-seq and why they chose mTORC1 pathway as the downstream mechanism.

REVIEWER COMMENTS

Reviewer #1 (Remarks to the Author):

The authors investigated the role of TRMT6 in HSC, and found that deletion of TRMT6 in hematopoietic cells via Mx1-Cre resulted in defective hematopoiesis and reduced repopulating potential of HPSC. Mechanistic studies revealed hyperactivation of mTORC1 as the cause of HSC defects. The experiments were well-performed and most conclusions are appropriate. However, a key weakness of this study is the lack of mechanism of TRMT6 regulation of TSC1.

Response: Thank you for your nice comments on our article. According to your suggestions, we have supplemented several data here and corrected several mistakes in our previous draft. Based on your comments, we also attached a point-by-point letter to you and the other two reviewers. We have made extensive revisions to our previous draft. The detailed point-by-point responses are listed below.

1. Line 30, in the abstract, it was mentioned that “Trmt6-null mice became moribund due to haematopoietic failure with pancytopenia in the blood and bone marrow 4 to 8 weeks after Trmt6 deletion”. The authors did not provide data about these conclusions, such as survival curve of KO mice. And, “pancytopenia” was not evident from Fig. S1H-S1L, it seems that KO mice have selective reduction of B cells.

Response: Thanks for the reviewer's kind suggestion. We have provided the survival curve of *Trmt6* cKO mice and the evidence for hematopoietic failure 4 weeks after *Trmt6* deletion (Please see the revised Figure S2K and S4F) to support our results.

2. Line 118, "Those results suggested a bias in the lineage output towards to myeloid lineages" is not accurate. In Fig. S1K, T cells in PB from KO mice were increased. Thus, there was a selective reduction of B cells, but not T cells, which does not support myeloid bias. Absolute cell number of T, B and myeloid cells should be provided.

Response: We have provided the absolute cell number of PB cells in *Trmt6* cKO mice 1 weeks after *Trmt6* deletion (Please see the revised Figure S2F-I).

"Those results suggested a bias in the lineage output towards to myeloid lineages" has been revised to "Those results suggested a dysfunction in the lineage output".

3. Whether the phenotypes were dependent on enzyme activity of TRMT6 was not addressed. Rescue experiments with WT TRMT6 and enzyme activity defective mutant of TRMT6 should be performed to answer the question. It is possible that TRMT6 might regulate HSC independent of RNA modification.

Response: To further determine whether TRMT6 exerted function through its

enzymatic activity, we generated TRMT6^{R377L} mutant construct. Since R377 of TRMT6 is essential for its binding with tRNA backbone, mutation of this site abolished the enzymatic activity. WT TRMT6 but not TRMT6^{R377L} mutant (Mut) overexpression in TRMT6 depleted HSCs could rescue HSC transplantation (Figure S8A-C), indicating that TRMT6 directly regulate HSC function dependent on its enzyme activity.

4. Line 228, “TSC1 were significantly upregulated in *Trmt6*-null LT-HSCs and LSK cells (Figures 3E and S4B)”. TSC1 was downregulated, not upregulated.

Response: We would like to thank the reviewer for pointing this out. TSC1 was indeed downregulated, and this sentence has been corrected as “TSC1 were significantly downregulated in *Trmt6*-null LT-HSCs and LSK cells”. (Page 12, Line13)

Also, the level of TSC1 should be further confirmed by WB.

Response: The decreased protein level of TSC1 was confirmed by WB as suggested by reviewer (Figure S6A).

Partial reduction of TSC1 is unlikely to cause a dramatic increase of pS6, since TSC1^{+/-} mice do not have obvious increase of pS6 and defects in hematopoiesis. Does TRMT6-KO HSC had other features of mTORC1 hyperactivation, such as increase cell size? Increase ROS? Whether

overexpression of TSC1 could rescue the KO phenotypes? These are weak points of this study.

Response: Thanks for the reviewer's kind suggestion. We feel sorry that we did not provide enough information about TRMT6-KO HSC previously. TRMT6-KO HSC also had other features of mTORC1 hyperactivation, such as increased cell size (Figure S6B) and increased ROS (Figure S6C). We have followed the reviewer's suggestion and included the overexpression of TSC1 in TRMT6-KO HSC analysis in the revised Figure S8. We find that overexpression of TSC1 could partially rescue the KO phenotypes. This evidence just suggests that TRMT6-TSC1-mTOR axis could be prominent for HSC function.

5. How TRMT6 specifically regulates translation of *Tsc1* but not other mRNA should be explored or discussed.

Response: Thanks for the reviewer's kind suggestion. We performed tRNA-m¹A-seq and Ribo-seq assays to study translation of gene in TRMT6-KO HSPC. The results showed that translation of *Tsc1* is significant difference in TRMT6-KO HSPC. (Please see the revised Figure 3C, S5 and S8)

6. The phenotype of TRMT6-KO is most close to TSC1/SZT2 KO mice reported recently, which should be discussed.

Response: Thanks for the reviewer's kind suggestion. We refer this

important paper (JCI--"SZT2 maintains hematopoietic stem cell homeostasis via nutrient-mediated mTORC1 regulation"). This study reported that constitutive nutrient signaling to mTORC1 in the absence of SZT2 was detrimental to the repopulating capacity of the HSCs⁶⁶. We found that the phenotype of SZT2 deficiency in HSCs is similar with Trmt6-deficient HSCs. So How SZT2, TSC1, PTEN and TRMT6 synergistically regulate mTORC1 signaling in HSCs requires further investigation as well.

Reviewer #2 (Remarks to the Author):

The manuscript “tRNA m1A modification regulate HSC maintenance and self-renewal via mTORC1 signaling” aims at examining how the m1A modification impacts hematopoiesis. The authors find TRMT6 (the binding component of the m1A methyltransferase complex) is upregulated following transplantation, and treatment of HSCs with ionizing radiation, LPS etc. To begin to understand the role of TRMT6-mediated m1A modification in hematopoiesis, the authors generated a *Trmt6* conditional knockout mouse, and determined that the loss of *Trmt6* impacts HSC quiescence and self-renewal. The authors also show that the mTORC1 signaling pathway, which is essential for HSC maintenance and self-renewal, is upregulated in HSC-enriched cell populations following *Trmt6* deletion. The authors then try to link loss of *Trmt6*, with reduced m1A modification in tRNAs, and suggest that this results in less efficiently translated TSC1 mRNA (a negative regulator of mTOR).

1. My major concern is with the lack of detail in the proposed mechanism. The rescue experiments in Figure 4 show that the mTORC1 pathway, can ameliorate the defects in self-renewal and proliferation of HSCs, following loss of *Trmt6*. However, the mechanism is not at all convincing without the addition of

m1A-tRNA-seq analysis, RiboTag, and differential protein analysis. Essentially the authors claim that loss of *Trmt6* results in less m1A tRNA modification which in turn specifically results in reduced translation of TSC1, and in turn aberrant mTOR signaling and deficiencies in homeostasis and regeneration of HSCs. Can the authors specifically show reduction of m1A modification on specific tRNAs?

Response: We understand the Reviewer's concern and agree with the reviewer's comment that the mechanism is not convincing with the results in the initial submission, therefore following the reviewer's suggestion, we conducted m¹A-tRNA-seq analysis, Ribo-seq, and differential protein analysis as described below. We performed mim-tRNA-seq (method from Molecular cell 2021) on the *Trmt6*^{+/+} and *Trmt6*^{-/-} HSPCs, and found that m¹A58 of many tRNA were specifically reduced in *Trmt6*^{-/-} versus *Trmt6*^{+/+} HSPCs (Please see the revised Figure S5 and 3C).

2. The authors have not mentioned how much RNA they isolate. But as a little as 200 ng of small RNAs could be used for **m1A-seq**. The authors should check how tRNA expression levels change following *Trmt6* deletion. Authors should specifically determine how the tRNA pool changes in terms of expression and/or modification levels following *Trmt6* deletion. The authors will need to check

whether loss of m1A modification is related to translation initiation or elongation. It could be an initiation effect. The m1A modification is key to stability of the initiation tRNA. The authors may need to differentiate initiation from elongation effects.

Following *Trmt6* deletion, the authors should check for the enrichment of hypo-m1A modified tRNA codons in *TSC1* compared to other genes? If this is the case, it would be nice to see a codon switch experiment to confirm this. However, considering m1A is found on many tRNA species, the authors should determine the whole set of mRNAs whose translation efficiency is affected. To this end, authors should cross a RiboTag mouse with their Cre mouse and interrogate ribosome-associated transcriptomes in their system at a global level. Following that, the authors should confirm differential protein expression. Authors should at least show differential protein expression of *TSC1*.

Response: About 500 ng of small RNA isolated from HSPCs was used for mim-tRNAseq. Ala tRNA were significantly dysregulated at expression level following *Trmt6* deletion. To determine whether *TSC1* translation deficiency was due to mRNA translation initiation or elongation in *Trmt6*^{-/-} HSPCs, we analyzed the m¹A58 levels and expression levels of initiator-methionine tRNA and found that the modification but not the expression level was decreased in

Trmt6^{-/-} HSPCs (Please see the revised Figure 3C, S5 and S8), suggesting that tRNA-m¹A58 is likely to regulate the translation elongation of *Tsc1* mRNA. Western blot confirmed that the protein expression of TSC1 decreased (Figure S6A).

3. In addition to TRMT6 what other genes involved in translation change upon transplantation, LPS etc? Can the authors compare changes of TRMT6 with other genes? A figure showing the most significant changes in genes including transcriptional factors and translation components would be good. Along those lines, did the authors look at TRMT61A expression changes upon transplantation, radiotherapy, chemotherapy, and inflammatory stimuli? Do other 'writers', 'readers' and 'erasers' change expression?

Response: We checked the expression changes of those genes the reviewer mentioned upon transplantation, radiotherapy, chemotherapy, and inflammatory stimuli, and found that: 1) both genes involved in translation and those involved in transcription were more likely to be up-regulated upon transplantation, radiotherapy, chemotherapy, and inflammatory stimuli, and those involved in translation likely to be affected in a higher degree, 2) the expression of *Trmt6* was up-regulated at a higher level than the average level of translation-related genes, and 3) the expression of the 'writers', 'readers' and 'erasers' of m¹A modification including *Trmt61a*, changed in different directions and at variable levels upon transplantation, radiotherapy, chemotherapy, and inflammatory stimuli, of those, *Trmt6* was always among

the top 3 up-regulated m¹A regulators (see the figure below).

4. Considering m⁶A has been shown to be preferentially required for adult HSC differentiation in vivo and in vitro, the authors should check that TSC1 is not marked by m⁶A, seeing that m⁶A is known to affect translation.

Response: Following the reviewer's suggestion, we checked if *Tsc1* was marked by m⁶A using the SLIM-seq data published by others, and found that m⁶A marked segments were enriched in *Tsc2* but not in *Tsc1* (see the figure below), suggesting the translation of *Tsc1* is less likely regulated through m⁶A modification. By the way, all other mTOR signaling-related genes shown in Figure 3F were also not marked by m⁶A.

Minor points:

1. Figure 1: B) Mix Cre should be Mx1-Cre

Response: We appreciate the reviewer's suggestion, and have replaced all the "Mix Cre" with "Mx1-Cre".

2. Page 12 - Line 229 do the authors mean upregulated in *Trmt6*-null or down regulated?

Response: We would like to thank the Reviewer for pointing this out. TSC1 was indeed downregulated, and this sentence has been corrected as "TSC1 were significantly downregulated in *Trmt6*-null LT-HSCs and LSK cells". (Page 12, Line13)

3. Page 13 – I would be more careful with the wording used in section "Inhibition of mTOR pathway ameliorates aberrant proliferation and self-renewal defects of *Trmt6* knockout HSC". For

example, Line 247 – 249. There is a decrease in p-S6 levels in Trm6 KO mouse after rapamycin treatment compared to vehicle (it is not a decrease compared to Trmt6 WT mouse). Or just say levels of p-S6 are rescued following rapamycin treatment in Trmt6 knockout mouse.

Response: Following the reviewer's suggestion, we changed the sentence "rapamycin treatment tended to reduce the p-S6 levels in LT-HSCs and LSK cells in *Trmt6*^{-/-} mice compared with that in *Trmt6*^{+/+} (Figures 4E and S4H)" to "There is a decrease in p-S6 levels in *Trmt6*^{-/-} mouse after rapamycin treatment compared to vehicle, but no changes in *Trmt6*^{+/+} mouse, suggesting that rapamycin treatment partially rescued the overactivation of mTOR signaling". We also carefully rewrite the whole section as highlighted in red in the revised manuscript.

4. Authors should include amounts of RNA used in the methods section.

Response: We included the amounts of RNA used in experiments in the methods section.

Reviewer #3 (Remarks to the Author):

In this study, Zuo H et al studied the roles and mechanism of Trmt6 in regulating HSC maintenance and self-renewal. Using a conditional Trmt6 KO mouse model, they found that Trmt6 KO resulted in the transition of quiescent HSCs into the cell cycle, increased HSPC frequencies and decreased reconstitution potential. Mechanistically, Trmt6 KO led to translational repression of Tsc1, which activated mTORC1 signaling pathway and led to HSC exhaustion. Finally, they found rapamycin ameliorates aberrant proliferation and self-renewal defects of Trmt6 KO HSCs. Overall, the role and regulatory mechanisms of Trmt6 in adult HSC maintenance are still largely unknown. The study is carefully designed, and the manuscript is well written. However, this study is rather preliminary with a lot of phenotypic data, but lacks comprehensive mechanistic investigation. Several issues need to be addressed.

Response: We would like to thank the Reviewer's positive comments on our study design and manuscript writing, and also fully understand the Reviewer's concern about the lack of comprehensive mechanistic investigation. By following the Reviewer's useful suggestions, we have enhanced the mechanistic investigation, please see below in details.

Major issues:

1. The authors claims the importance of tRNA m¹A modification in the regulation of HSC maintenance and self-renewal. Given *Trmt6* and *Trmt61a* form m¹A methyltransferase complex, it would strengthen their conclusion by doing these analyses.

(1) To examine the expression of *Trmt6* gene across multiple hematopoietic cell populations utilizing publicly available single-cell transcriptomic data.

Response: Thanks for the reviewer's kind suggestion. In Fig. S1 of the initial submission, we included the expression of *Trmt6* gene across multiple hematopoietic cell populations utilizing publicly available single-cell transcriptomic data. While as the reviewer saying, *Trmt6* and *Trmt61a* form m¹A methyltransferase complex, thus we additionally checked the expression of *Trmt61a*, and found that *Trmt61a* expressed in HSPCs, although at a lower level than *Trmt6* (Figure S1).

(2) To check whether *Trmt61a* KO phenocopies the HSC defects in *Trmt6* KO mice. To demonstrate whether the tRNA m¹A-TSC1-mTORC1 axis is also responsible for *Trmt61a* KO HSCs.

Response: We agree that more study would be useful to understand details of *Trmt6-Trmt61a* complex function in HSCs. At this point we do not have the

Trmt61a^{fllox} mice to study the *Trmt61a* cKO phenotype. So we have investigated *Trmt61a* function in HSC by using Lentivirus-shRNA_of mouse *Trmt61a*. The main findings are presented in Figure S7 and reveals TRMT61A could be play a similar function with TRMT6 in HSCs.

We found that tRNA m¹A level and TSC1 protein level was decreased in sh*Trmt61a*-HSCs compared with that in the control HSCs (Figure S7a and Figure S7F, S7G). And pS6 levels was increased in sh*Trmt61a*-HSCs compared with that in the control HSCs (Figure S7E). These results revealed tRNA m¹A-TSC1-mTORC1 axis could be also responsible for *Trmt61a*-knockdown HSCs.

2. In Fig. S1, the authors showed that *Trmt6* was dysregulated in HSC upon different treatment or during aging. However, they only examined the effect of *Trmt6* in steady hematopoiesis. To further consolidate their data, they should check the effect of *Trmt6* on HSC number and functions under stress conditions, such as 5-Fu, irradiation and aging.

Response: We want to check the effect of *Trmt6* on HSC number and functions under stress conditions, such as 5-Fu, irradiation. But the KO mice is die quickly after 5-Fu, irradiation (Figure S3B-E). These results showed that *Trmt6* is more important in stress hematopoiesis.

3. In Fig.1, the authors performed single-cell RNA sequencing (scRNA-seq) on sorted LSK cells isolated from bone marrow (BM) of young adult Trmt6+/+ and Trmt6-/- mice. However, the UMAP clustering analysis in Fig. 1C is not convincing because most of the HSPC clusters are hard to distinguish, particularly for LT-HSC, ST-HSC and MPPs. The authors should check the expression of known markers for these populations, such as Sca1, c-Kit, CD34, FLK2, CD150, CD48, CD127, etc.

Response: We understand the reviewer's concern. Because we annotated cells on the basis of transcriptome similarity between our cells and the cells sorted using specific surface markers of HSPCs, the resulted different populations of HSPCs are indeed not that easy to distinguish. Meanwhile, when we checked the expression of known markers as the reviewer suggested, and found they are consistent with the cell annotation (see figure below).

4. In Fig.2, the authors conducted competitive transplantation experiments by transplanting the same amount of total bone marrow cells and found that *Trmt6* deletion produces a cell-autonomous functional defect in HSCs. Since the frequency of LT-HSCs is increased in WT and *Trmt6* KO mice, it would be better to sort the LT-HSC and transplant with equal number of LT-HSC.

Response: We also conducted equal number of HSC competitive transplantation experiments. Surprisingly, in comparison with *Trmt6*^{+/+} HSCs, *Trmt6*^{-/-} HSCs exhibited a ~10-fold reduction in their long-term repopulating ability followed by PB and LT-HSC chimerism analysis (Figure S3M and S3N).

5. In the whole study, the authors evaluated the effect of *Trmt6* on HSCs by use of *Mx1-cre*; *Trmt6* floxed mice. They gave the mice with 6 times of poly I:C injections and performed the experiments and scRNA-seq only 1 week after injection. In the HSC field, many groups usually examined the HSC phenotype 3-4 weeks after 6 times of poly I:C injections so that the genetic deletion would be more complete. Considering that *Trmt6* was not completely deleted in LSK cells (Fig. S1G), they should check the HSC phenotype after a longer time (over 4 weeks after poly I:C injection) to evaluate whether the effect was stronger.

Response: We check the HSC phenotype at 4 weeks after poly I:C injection

to evaluate the effect of *Trmt6* deficiency in HSCs. We have provided the survival curve of *Trmt6* cKO mice and the evidence for hematopoietic failure 4 weeks after *Trmt6* deletion (Please see the revised Figure S2K and S4F) to support our results.

6. In Fig.2B-D, the authors showed that the frequencies of HSPCs and committed progenitors were all increased. Besides, they showed the frequency of B cells was reduced whereas the frequency of myeloid cells was augmented (Fig. S1J-K). However, the bone marrow cellularity was declined in Fig. S1L. Could the authors explain why the total bone marrow cell number was dramatically decreased in the context of HSPC expansion?

Response: We detected BM cell apoptosis. Maybe cell apoptosis leads to the decreased in total bone marrow cell number (Please see the revised Figure S6E).

7. Given that *Trmt6* KO led to activation of mTORC1 signaling pathway and HSC exhaustion, whether apoptosis rate and ROS levels are also increased in the *Trmt6* KO HSCs? And whether mitochondrial metabolism is also altered in the *Trmt6* KO HSCs?

Response: Thanks for the reviewer's kind suggestion. We feel sorry that we did not provide enough information about TRMT6-KO HSC previously.

TRMT6-KO HSC also had other features of mTORC1 hyperactivation, such as increased cell size (Figure S6B) and increased ROS (Figure S6C), peak mitochondrial membrane potential changes (Figure S6D). We have followed the reviewer's suggestion and included the overexpression of TSC1 in TRMT6-KO HSC analysis in the revised Figure S8. We find that overexpression of TSC1 could partially rescue the KO phenotypes. This evidence just suggests that TRMT6-TSC1-mTOR axis could be prominent for HSC function.

8. In Fig.3, they found that the protein levels of the upstream targets (LKB1, AMPK, GSK3 β , REDD1 and RHEB) of mTORC1 pathways did not differ (Figure S4C), but TSC1 were significantly downregulated in Trmt6-null LT-HSCs and LSK cells (Figures 3E and S4B). Besides, they performed polyribosome RT-PCR experiments to quantify the ribosome occupancy of Tsc1 mRNAs, and found a dramatic decrease in the accumulation of ribosomes on Tsc1 mRNAs but not on LKB1, AMPK, GSK3 β , REDD1, RHEB or control transcripts upon TRMT6 depletion. This is quite interesting, since it looks like Trmt6 specifically regulated the translation of Tsc1. How is the specificity executed by TRMT6? The authors should provide more mechanistic investigation into these interesting findings.

Response: We understand the Reviewer's concern and agree with the reviewer's comment that the mechanism is not convincing with the results in the initial submission, therefore following the reviewer's suggestion, we conducted m¹A-tRNA-seq analysis, Ribo-seq, and differential protein analysis as described below. We performed mim-tRNA-seq (method from Molecular cell 2021) on the *Trmt6*^{+/+} and *Trmt6*^{-/-} HSPCs, and found that m¹A58 of many tRNA were specifically reduced in *Trmt6*^{-/-} versus *Trmt6*^{+/+} HSPCs (Please see the revised Figure S5 and 3C).

Minor issues:

1. Beyond the flow cytometry, Hemogram analysis should be performed in the conditional *Trmt6* KO mice to evaluate the detailed information of all the blood cells.

Response: Hemogram analysis in Figure S2F-I.

2. Fig.2A and Fig.2D were not mentioned in the result session.

Response: Revised Fig.2A and Fig.2D were mentioned in the result session (Page 7, line13)

3. In the abstract, the authors mentioned that “*Trmt6*-null mice became moribund due to hematopoietic failure with pancytopenia in the blood and bone marrow 4 to 8 weeks after *Trmt6* deletion.”

However, there was no data on this conclusion in the manuscript.

Response: Thanks for the reviewer's kind suggestion. We have provided the survival curve of *Trmt6* cKO mice and the evidence for hematopoietic failure 4 weeks after *Trmt6* deletion (Please see the revised Figure S2K and S4F) to support our results.

4. The abstract should be re-written since a lot of useful information was not mentioned.

Response: The abstract have been re-written.

5. The authors showed the scRNA-seq data in the Fig.1. It might be better to present the phenotypic and transplantation results (Fig.2) in the Fig.1, and put the scRNA-seq data in the Fig.2. In this case, it would easier to understand why they needed to perform scRNA-seq and why they chose mTORC1 pathway as the downstream mechanism.

Response: We appreciate the reviewer's comment. We have made the change following the Reviewer's suggestion, and changed the order of supplemental figures and text related to those two figures correspondingly.

REVIEWER COMMENTS

Reviewer #1 (Remarks to the Author):

Most responses are satisfactory, except for this one:

3. Whether the phenotypes were dependent on enzyme activity of TRMT6 was not addressed. Rescue experiments with WT TRMT6 and enzyme activity defective mutant of TRMT6 should be performed to answer the question. It is possible that TRMT6 might regulate HSC independent of RNA modification.

Response: To further determine whether TRMT6 exerted function through its enzymatic activity, we generated TRMT6R377L mutant construct. Since R377 of TRMT6 is essential for its binding with tRNA backbone, mutation of this site abolished the enzymatic activity. WT TRMT6 but not TRMT6R377L mutant (Mut) overexpression in TRMT6 depleted HSCs could rescue HSC transplantation (Figure S8A-C), indicating that TRMT6 directly regulate HSC function dependent on its enzyme activity.

Concerns: this experiment (Fig. S8A-C) lack an important control: rescue of the phenotype of by wild-TRMT6.

Reviewer #2 (Remarks to the Author):

I appreciate the authors performing the mim-tRNAseq and Riboseq experiments. The authors claim that most tRNAs have less m1A58, however in Fig S5 C, the heat maps of WT and TRMT6 -/- look very similar; yet Fig 3 suggests more widespread dysregulation. The volcano plot in Fig 5S F looks strange and needs to be addressed. TSC1 and the upstream targets of mTORC1 pathway should be labeled in this volcano plot. More importantly, the authors don't provide any direct evidence that the dysregulation in tRNA Ala is the reason that decreased translation efficiency of TSC1 is observed.

Line 254, what are the authors referring to when they say 'elevated tRNAs'? Was tRNA expression measured?

Line 259, the authors say that expression of initiator tRNA doesn't change. Again neither of the figures mentioned show tRNA expression. It doesn't seem like tRNA expression was measured.

It is good to see that Trmt6 -/- HSC also had other features of mTORC1 hyperactivation (Fig S6).

Check Fig S8. B and C I do not see rescue after overexpression of WT TRMT6 in *Trmt6* ^{-/-}. Check labels of S8F, should be OE-TSC1 not OE-Trmt6/mut, right?

Also the placement of the newly added text needs to be better thought out. The manuscript doesn't flow well.

Reviewer #3 (Remarks to the Author):

I'm satisfied with the authors responses to my previous comments and suggestions. However, a recent study has reported the role and mechanism for the TRMT6-TRMT61A complex in HSC aging (Hanqing He et al. *Nat Aging*. 2024 Jan 17). The authors should discuss this paper in the Introduction and Discussion.

REVIEWER COMMENTS

Reviewer #1 :

Most responses are satisfactory, except for this one:

3. Whether the phenotypes were dependent on enzyme activity of TRMT6 was not addressed. Rescue experiments with WT TRMT6 and enzyme activity defective mutant of TRMT6 should be performed to answer the question. It is possible that TRMT6 might regulate HSC independent of RNA modification. Response: To further determine whether TRMT6 exerted function through its enzymatic activity, we generated TRMT6 R377L mutant construct. Since R377 of TRMT6 is essential for its binding with tRNA backbone, mutation of this site abolished the enzymatic activity. WT TRMT6 but not TRMT6R377L mutant (Mut) overexpression in TRMT6 depleted HSCs could rescue HSC transplantation (Figure S8A-C), indicating that TRMT6 directly regulate HSC function dependent on its enzyme activity. **Concerns: this experiment (Fig. S8A-C) lack an important control: rescue of the phenotype of by wild-TRMT6.**

Response: Thank the reviewer for figuring it out, we have added the control (rescue of the phenotype of by wild-TRMT6) in this experiment following your suggestion (See the figure below and Figures S8D-F in the revised manuscript).

Reviewer #2 (Remarks to the Author):

I appreciate the authors performing the mim-tRNAseq and Riboseq experiments. The authors claim that most tRNAs have less m¹A58, however in Fig S5 C, the heat maps of WT and TRMT6^{-/-} look very similar; yet Fig 3 suggests more widespread dysregulation.

Response: Thanks for your comments on the result of mim-tRNAseq, and we understand the reviewer's concern. The Figures 3C indeed showed that most tRNAs have less m¹A58, this observation is quite similar to the effects of TRMT61A depletion in tumor and T cell (Liu et al., 2022; Wang et al., 2021). However, in the heatmap, there is a wide range of modification levels of tRNAs indicated by the colors, so that only several of these tRNAs showed changes in modification proportions large enough to be noticeable by comparing the colors (Fig 3), especially when the modification levels at position 58 for each WT and TRMT6^{-/-} were not placed side by side. Of note, the tRNAs are clustered independently in the heatmaps for WT and TRMT6^{-/-}, which is the other reason why these two heatmaps look very similar.

Liu, Y., Zhou, J., Li, X., Zhang, X., Shi, J., Wang, X., Li, H., Miao, S., Chen, H., He, X., *et al.* (2022). tRNA-m(1)A modification promotes T cell expansion via efficient MYC protein synthesis. *Nat Immunol* 23, 1433-1444.

Wang, Y., Wang, J., Li, X., Xiong, X., Wang, J., Zhou, Z., Zhu, X., Gu, Y., Dominissini, D., He, L., *et al.* (2021). N(1)-methyladenosine methylation in tRNA drives liver tumorigenesis by regulating cholesterol metabolism. *Nat Commun* 12, 6314.

The volcano plot in Fig 5S F looks strange and needs to be addressed. TSC1 and the upstream targets of mTORC1 pathway should be labeled in this volcano plot.

Response: Thanks for the reviewer's comment and suggestion. As suggested by the reviewer, we labeled genes encoding TSC1 and the upstream targets of mTORC1 pathway on the new volcano plot (see the figure below and Figure S6F in the revised manuscript). As shown in the figure, the translation efficiency of TSC1 but not the upstream targets of mTORC1 pathway were affected by TRMT6 depletion, consistent with previous results in Fig 3F-H.

More importantly, the authors don't provide any direct evidence that the dysregulation in tRNA Ala is the reason that decreased translation efficiency of TSC1 is observed.

Response: Thank you for emphasizing the importance of additional experiments. Recently, we provide the direct evidence by replacing the codon Ala-GCA and Ala-GCG (corresponding to the two most affected tRNA Ala-TGC and Ala-CGC, respectively) in TSC1 by Ala-GCC (see the figure A below and Figure S7A in the revised manuscript), because the modification of tRNA Ala-GGC was too low to be detected, thus less affected by TRMT6 depletion. Expression of this mutant *Tsc1* via lentivirus in WT and *Trmt6*-deficient HSPCs was sufficient to rescue the defective expression of TSC1 and single clone formation in KO cells *in vitro* (see the figure below and Figure S7 in the revised manuscript), confirming that tRNA-m¹A58 modification directly regulates *Tsc1* mRNA translation through codon decoding.

Line 254, what are the authors referring to when they say 'elevated tRNAs'? Was tRNA expression measured? Line 259, the authors say that expression of initiator tRNA doesn't change. Again neither of the figures mentioned show tRNA expression. It doesn't seem like tRNA expression was measured.

Response: Thank you for the comments. For the first question, the 'elevated tRNAs' should be 'tRNAs', sorry for making it confusing by the clerical error. For the second, the answer is: Yes, both the expression and modification level of tRNAs were measured by mim-tRNA-seq. Nedialkova Lab developed the mim-tRNA-seq method to measure the expression and modification level of tRNAs at the same time, through the efficient reverse transcription of highly modified RNAs during library construction using TGIRT and the efficient and accurate mapping and analysis of tRNA-derived reads with single-transcript resolution by multiple conceptual advances in the analysis of tRNA sequencing data, including the use of modification

annotations. In our manuscript, the scatter plot in Figure S5B (Figure S6B in the revised manuscript) showed the tRNA expression $Trmt6^{+/+}$ and $Trmt6^{-/-}$ HSPCs. We feel sorry for missing the reference of the figure illustrating the mentioned expression change, and added the reference of the figure (Figure S6B) in the text (Page 13 Line 22).

It is good to see that $Trmt6^{-/-}$ HSC also had other features of mTORC1 hyperactivation (Fig S6). Check Fig S8. B and C I do not see rescue after overexpression of WT TRMT6 in $Trmt6^{-/-}$.

Response: Thank the reviewer for figuring it out, we have added the control (rescue of the phenotype of by wild-TRMT6) in this experiment following your suggestion (See the figure below and Figures S8D-F in the revised manuscript).

Check labels of S8F, should be OE-TSC1 not OE- $Trmt6$ /mut, right?

Response: We would like to thank the Reviewer for pointing this out. It should be OE-TSC1 (See the figure below and Figure S10C in the revised manuscript).

Also the placement of the newly added text needs to be better thought out. The manuscript doesn't flow well.

Response: Thank for the reviewer's comment, we have rewritten the parts that don't go so well, with a particular focus on newly added text.

Reviewer #3 (Remarks to the Author):

I'm satisfied with the authors responses to my previous comments and suggestions. However, a recent study has reported the role and mechanism for the TRMT6-TRMT61A complex in HSC aging (Hanqing He et al. Nat Aging. 2024 Jan 17). The authors should discuss this paper in the Introduction and Discussion.

Response: Thank for the reviewer's comment, we also noticed this paper (Hanqing He et al. Nat Aging. 2024 Jan 17) after the last revision, and discuss this paper about the role and mechanism for the TRMT6-TRMT61A complex in HSC aging in the Introduction (Page 4, Line15-17) and Discussion (Page 19, Line 5-9) in the revised manuscript.

REVIEWERS' COMMENTS

Reviewer #1 (Remarks to the Author):

All my concerns have been addressed. I do not have further comments.

Reviewer #2 (Remarks to the Author):

I am satisfied with the authors responses.

I noticed single is misspelt in Fig S7B.